# LATENT WAVELET DIFFUSION FOR ULTRA-HIGH-RESOLUTION IMAGE SYNTHESIS

**Luigi Sigillo**[1,2,3,*]**, Shengfeng He**[2]**, Danilo Comminiello**[1]

[1]Sapienza University of Rome, [2]Singapore Management University, [3]EMBL
`luigi.sigillo@uniroma1.it`

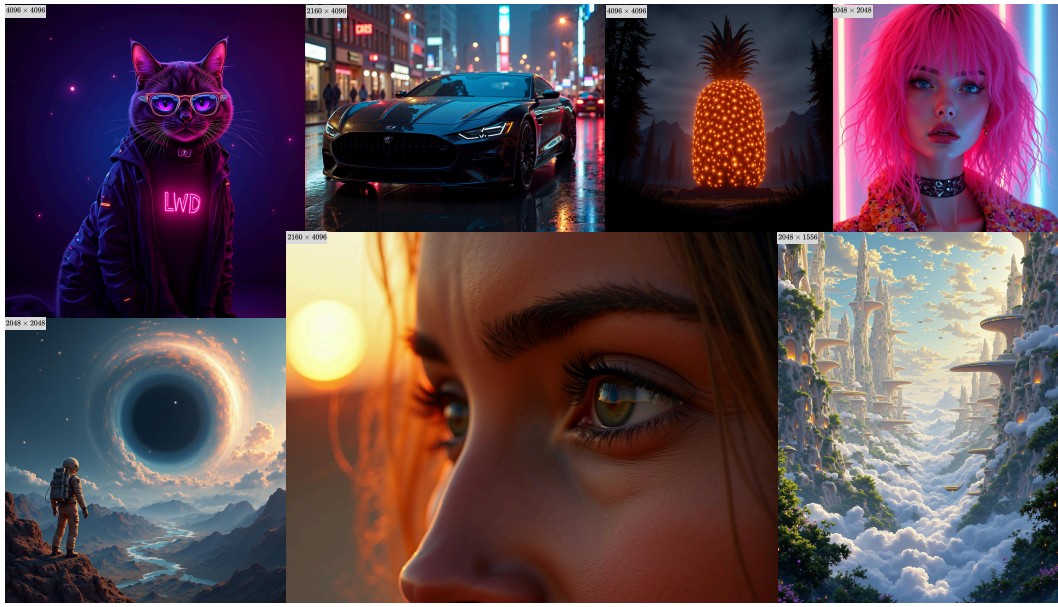

Figure 1: We propose Latent Wavelet Diffusion, achieving 4K image synthesis without architectural changes or additional inference cost to existing latent diffusion models.

## ABSTRACT

High-resolution image synthesis remains a core challenge in generative modeling, particularly in balancing computational efficiency with the preservation of fine-grained visual detail. We present *Latent Wavelet Diffusion (LWD)*, a lightweight training framework that significantly improves detail and texture fidelity in ultra-high-resolution (2K-4K) image synthesis. LWD introduces a novel, frequency-aware masking strategy derived from wavelet energy maps, which dynamically focuses the training process on detail-rich regions of the latent space. This is complemented by a scale-consistent VAE objective to ensure high spectral fidelity. The primary advantage of our approach is its efficiency: LWD requires no architectural modifications and adds zero additional cost during inference, making it a practical solution for scaling existing models. Across multiple strong baselines, LWD consistently improves perceptual quality and FID scores, demonstrating the power of signal-driven supervision as a principled and efficient path toward high-resolution generative modeling. The code is available at `https://github.com/LuigiSigillo/LatentWaveletDiffusion`.

## 1 INTRODUCTION

Diffusion models have become a dominant paradigm in generative modeling, achieving state-of-the-art results in tasks such as image synthesis (Podell et al., 2024; Sauer et al., 2024; Lopez et al., 2025; Wang et al., 2025), super-resolution (Saharia et al., 2022; Sigillo et al., 2024; Wang et al., 2024), and image editing (Brooks et al., 2023; Meng et al., 2022). Foundational methods like Denoising Diffusion Probabilistic Models (DDPM) Ho et al. (2020) and Denoising Diffusion Implicit Models

---

[*]Work done during a visiting period at SMU. Now at European Molecular Biology Laboratory.

(DDIM) Song et al. (2021a) have enabled increasingly powerful variants. Latent Diffusion Models (LDMs) Rombach et al. (2022) improve efficiency by operating in a learned latent space, while recent architectures such as Diffusion Transformers (DiTs) (Peebles & Xie, 2023; Esser et al., 2024) further enhance scalability and modeling capacity.

Despite recent advances, generating ultra-high-resolution (UHR) images at resolutions of 2K to 4K and beyond remains a significant challenge (Tragakis et al., 2024). Simply scaling models trained on lower-resolution data often fails to generalize, leading to repeated structures, blurred textures, and spatial inconsistencies (Bar-Tal et al., 2023; He et al., 2023). Naive alternatives, such as hierarchical generation pipelines and post-hoc super-resolution techniques (Du et al., 2024a; He et al., 2023), typically produce oversmoothed outputs. To overcome these limitations, several strategies have been explored. One direction involves direct UHR training or fine-tuning (Ren et al., 2024; Chen et al., 2024), although this typically demands extensive computational resources, access to proprietary high-resolution datasets (Liu et al., 2024b), and substantial GPU memory for large model backbones (Labs, 2024; Esser et al., 2024). Other efforts focus on architectural modifications to improve long-range dependency modeling (Liu et al., 2024a), or optimization techniques that enhance the quality of latent representations (Hahm et al., 2024).

Across these varied approaches, a common limitation persists: most methods treat all spatial regions uniformly during generation, applying the same refinement process to areas with vastly different visual complexity. This uniformity disregards local frequency variation, failing to distinguish between smooth regions and areas rich in textures, edges, or semantic structure. The consequences are twofold. First, computation is wasted on low-detail areas that require minimal refinement. Second, high-detail regions are not sufficiently supervised, leading to artifacts or loss of fidelity. These issues stem from both architectural and algorithmic constraints: latent representations often lack the structural granularity required for UHR synthesis, and current diffusion training objectives do not incorporate spatial adaptivity into the denoising process. Together, these limitations present a core bottleneck for perceptually accurate ultra-high-resolution image generation.

In this work, we propose *Latent Wavelet Diffusion* (LWD), a general and modular framework that introduces frequency-sensitive spatial supervision into the latent denoising process of diffusion models. LWD is motivated by the observation that different regions of an image exhibit varying levels of structural complexity and perceptual importance. While some areas contain intricate textures or sharp edges, others are homogeneous or low in detail. Our goal is to exploit this spatial heterogeneity by allocating greater learning signal to regions with high visual complexity, and reducing supervision in low-detail areas. Importantly, LWD achieves this adaptivity without modifying the underlying architecture of the diffusion model, making it broadly applicable across model families.

The LWD framework consists of three key components:

1. A **spectrally-aware VAE fine-tuning objective** that improves the structure and diffusability of latent representations. By encouraging semantic consistency and frequency regularity, this objective enhances the suitability of latent spaces for high-resolution generation. It serves as the foundation for the subsequent components of LWD.

2. A **wavelet-derived spatial saliency map**, computed via a discrete wavelet transform (DWT) applied to the latent representation. This map aggregates the local energy of high-frequency subbands (LH, HL, HH) and is normalized to highlight spatial regions with strong structural detail. These saliency maps are interpretable, data-driven, and require no additional training, providing a principled measure of spatial importance directly from the signal.

3. A **time-dependent masking strategy** that leverages the frequency-based saliency maps to modulate the training loss. At each spatial location and timestep, a binary mask determines whether the denoising loss is applied. Regions with high wavelet energy receive supervision across more timesteps, while simpler areas are updated less frequently. This mechanism introduces spatial adaptivity into the learning process and improves the fidelity of fine-scale detail.

LWD is compatible with a broad range of latent diffusion models, including both classical diffusion and flow-matching formulations. Because it operates solely through the training objective, LWD can be seamlessly integrated into existing pipelines. While ultra-high-resolution generation is inherently computationally demanding due to the backbone's scaling properties, LWD incurs zero marginal cost relative to the baseline. It requires no architectural changes or cascaded upsamplers, making it a practical solution for improving the generation quality of existing models. We demonstrate its

flexibility and effectiveness by applying it to several state-of-the-art latent diffusion models and evaluating performance on ultra-high-resolution image synthesis (2K to 4K). Experimental results show that LWD consistently enhances perceptual quality, improves FID scores, and better preserves fine-grained detail, all without increasing inference complexity.

## 2 RELATED WORK

**Diffusion Models and Latent Generation.** Diffusion models have become foundational in generative modeling, particularly for image synthesis (Shen et al., 2025; Zhan et al., 2025), by progressively denoising Gaussian noise using a learned score function. Variants based on stochastic differential equations (Song et al., 2021b), probability flow ordinary differential equations (Lipman et al., 2023), and reinforcement-trained objectives (Black et al., 2024) have expanded the design space with improved stability and sampling efficiency.

To reduce the cost of high-resolution generation, Latent Diffusion Models (LDMs) (Rombach et al., 2022) perform synthesis in a compressed space learned via variational autoencoders (VAEs). However, generation quality is closely tied to the spectral fidelity and structure of these latent representations. Prior work has sought to improve this through enhanced VAE architectures (Esser et al., 2021), hierarchical compression (Takida et al., 2024), and frequency-aware regularization (Skorokhodov et al., 2025; Kouzelis et al., 2025). We build on this direction by integrating frequency-sensitive supervision both during encoding and throughout the denoising process.

**Flow Matching and High-Resolution Diffusion.** Flow matching Lipman et al. (2023); Esser et al. (2024) offers an alternative to classical diffusion by learning a continuous velocity field that maps noise to data in latent space, eliminating the need for fixed noise schedules. This formulation underlies models such as Flux Labs (2024), which, paired with DiT backbones, has demonstrated strong performance in ultra-high-resolution pipelines (Zhang et al., 2025; Yu et al., 2025). Our method extends this family by introducing frequency-based spatial masking into the flow-matching objective. Through wavelet decomposition of the latent space, LWD computes spatial saliency maps that guide targeted supervision toward detail-rich regions, enhancing fine structure without modifying model architecture.

**Variational Autoencoder Optimization.** The performance of latent diffusion models at high resolutions depends critically on the expressiveness and spectral consistency of the VAE. Improvements include multi-scale encoders (Vahdat & Kautz, 2020; Takida et al., 2024), spectral loss functions (Björk et al., 2022), and scale-consistency constraints (Skorokhodov et al., 2025; Kouzelis et al., 2025). Wavelet-based methods (Esteves et al., 2025; Lin et al., 2024; Agarwal et al., 2025) enrich latent expressiveness by isolating frequency components, while compression-oriented approaches (Xie et al., 2025; Tang et al., 2024; HaCohen et al., 2024) aim to reduce token count for improved sampling efficiency. Our LWD fine-tunes a pretrained VAE with a scale-consistent spectral loss that suppresses spurious high-frequency noise. This regularized latent space facilitates downstream wavelet decomposition and supports our spatially adaptive denoising objective.

**Ultra High-Resolution Image Synthesis.** Maintaining global structure and fine detail in ultra-resolution generation is a persistent challenge. Standard diffusion models tend to produce repetitive patterns or distortions (He et al., 2023). Existing solutions include cascaded generation (Ho et al., 2022), progressive upsampling (Gu et al., 2024), domain-specific pipelines (Ren et al., 2024), and latent-space super-resolution (Jeong et al., 2025). However, training-based methods (Xie et al., 2023; Zheng et al., 2024; Guo et al., 2024; Chen et al., 2024) are resource-intensive, and training-free approaches (Bar-Tal et al., 2023; Lee et al., 2023) often yield local artifacts.

Methods such as ScaleCrafter He et al. (2023) mitigate repetition through dilated convolutions but may distort structure, while ResMaster Shi et al. (2025) uses low-resolution references for guided refinement. HiDiffusion Zhang et al. (2023) introduces architectural changes that risk performance trade-offs, and progressive strategies like DemoFusion Du et al. (2024a) suffer from slow inference and irregular patterns. Diffusion-4K Zhang et al. (2025) and URAE Yu et al. (2025) advance latent modeling and parameter-efficient adaptation at 4K resolution. Our LWD complements these approaches by introducing signal-driven, spatially adaptive supervision, which improves structural and perceptual fidelity at no additional cost.

**Generative Modeling in the Frequency Domain.** Frequency structure plays an increasingly important role in generative modeling (Yang et al., 2023). Wavelet-based diffusion methods, such

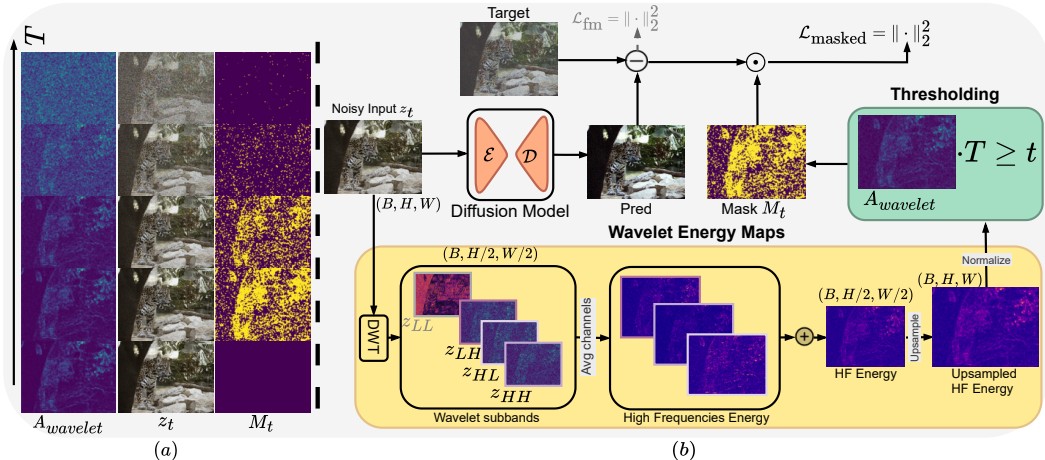

Figure 2: $(a)$ Temporal evolution of latent $z_t$, wavelet energy maps $A_{wavelet}$, and attention map $M_t$ across diffusion timesteps. $(b)$ Our wavelet-masked flow matching objective at a timestep $t$. The model computes a wavelet attention map $\mathbf{M}_t$ from latent $z_t$ to modulate the prediction error between target velocity field ($\epsilon - z_0$) and predicted velocity $v_\Theta(z_t, t, y)$. This focuses optimization on high-frequency regions with greater perceptual importance. While operations occur in latent space, decoded visualizations are shown for interpretability.

as WaveDiff Phung et al. (2023), and spectral decomposition approaches have proven useful for efficient sampling (Qian et al., 2024), super-resolution (Aloisi et al., 2026; Sigillo et al., 2025), and restoration (Huang et al., 2024b; Zhao et al., 2024; Jiang et al., 2023). FouriScale Huang et al. (2024a) demonstrated that frequency filtering enhances coherence, and DiffuseHigh Kim et al. (2025) leveraged low-frequency DWT guidance to improve global structure in UHR synthesis.

Diffusion-4K (Zhang et al., 2025) incorporated wavelet losses in the latent space to balance frequency bands, but applied them uniformly across all spatial locations. In contrast, LWD introduces wavelet-based spatial conditioning through time-dependent masking. Rather than treating frequency as a passive loss signal, our LWD actively uses local frequency energy to modulate supervision across space and time, concentrating on learning where detail matters most. This enables sharper, more coherent synthesis without modifying the underlying model or increasing inference cost.

## 3 METHODOLOGY

*Latent Wavelet Diffusion (LWD)* introduces frequency-aware supervision into latent diffusion models by coupling signal-driven saliency analysis with adaptive training. Our key insight is that structural complexity in images is unevenly distributed in space, yet most denoising models refine all positions equally. LWD addresses this by modulating the supervision schedule based on local frequency content, improving detail fidelity without increasing computational cost.

LWD operates in two sequential stages. The first stage fine-tunes a variational autoencoder (VAE) using a scale-consistency objective. This independent step yields spectrally stable latent representations, preparing the space for the subsequent frequency-based modulation.

In the second stage, we fine-tune a latent diffusion model (e.g., Flux) using a modified flow-matching objective that integrates frequency-guided supervision. This stage incorporates three tightly coupled components: (1) extraction of wavelet-based spatial saliency maps from latent codes; (2) construction of a time-dependent mask that adapts the training signal based on local frequency energy; and (3) application of this mask to modulate the training loss dynamically across spatial positions and timesteps. Together, these mechanisms introduce spatial adaptivity into the denoising process, directing learning resources toward detail-rich regions. All components are model-agnostic and can be integrated into standard latent diffusion pipelines without architectural changes.

### 3.1 VAE FINE-TUNING WITH SCALE-CONSISTENCY LOSS

High-resolution generation places unique demands on the latent space: it must retain both semantic structure and spectral coherence across scales. To ensure this, we fine-tune the variational autoencoder

(VAE) using a multi-resolution reconstruction objective that regularizes frequency content while preserving perceptual fidelity.

Formally, let $z = E(x)$ be the latent encoding of image $x$, $x_{\text{down}}$ a downsampled version of $x$, and $z_{\text{down}}$ the downsampled version of the latent $z$. Our loss combines four components:

$$\mathcal{L}_{\text{VAE}} = \underbrace{\|D(z) - x\|_2^2}_{\text{Reconstruction}} + \alpha \underbrace{\|D(E(z_{\text{down}})) - x_{\text{down}}\|_2^2}_{\text{Scale Consistency}} + \beta \underbrace{D_{\text{KL}}(q(z \mid x) \parallel p(z))}_{\text{Latent Regularization}} + \lambda \underbrace{\mathcal{L}_{\text{LPIPS}}(D(z), x)}_{\text{Perceptual Loss}}, \tag{1}$$

Here, we incorporate a scale-consistency term (Skorokhodov et al., 2025; Kouzelis et al., 2025) that encourages the VAE to maintain structural coherence across resolution scales. While originally proposed for general reconstruction, we identify it as critical for wavelet-guided UHR synthesis. Without this regularization, standard VAEs exhibit spurious high-frequency noise that confounds downstream wavelet masking. This preprocessing naturally suppresses compression artifacts while preserving essential structural information in $z$, enabling our masking strategy to target meaningful details rather than noise.

Unlike recent approaches that inject frequency conditioning directly into the encoder (Aloisi et al., 2026; Zhang et al., 2025), our formulation decouples signal regularization and generation: we first sculpt the latent space to exhibit desirable

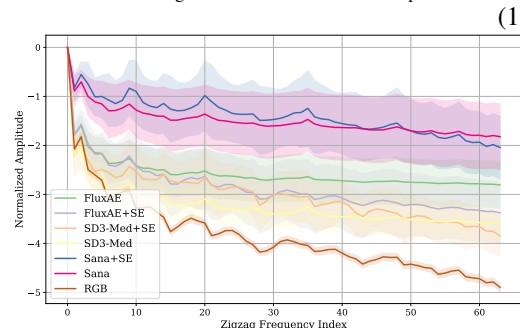

Figure 3: Normalized Discrete Cosine Transform amplitudes over zigzag frequency indices. The 'RGB' curve represents the target spectrum of real images. Our tuning (+SE) suppresses high-frequency energy corresponding to artifacts, aligning the latent spectrum with the RGB reference to ensure a cleaner latent space.

frequency properties, and then use this structure to guide the training of the diffusion model. This preserves architectural modularity while enabling effective frequency-aware supervision.

## 3.2 WAVELET-DERIVED FREQUENCY SALIENCY MAPS

To guide spatial supervision based on structural complexity, we extract saliency maps from latent representations using localized frequency analysis. Given a latent tensor $z \in \mathbb{R}^{C \times H \times W}$, we apply a single-level Discrete Wavelet Transform (DWT), producing four subbands:

$$\text{DWT}(z) \rightarrow \{z_{LL}, z_{LH}, z_{HL}, z_{HH}\}, \tag{2}$$

where $z_{LL}$ captures low-frequency approximations and $\{z_{LH}, z_{HL}, z_{HH}\}$ encode directional high-frequency detail.

We compute the localized high-frequency energy as:

$$E(i, j) = \frac{1}{C} \sum_c \left[ (z_{LH}^{c,i,j})^2 + (z_{HL}^{c,i,j})^2 + (z_{HH}^{c,i,j})^2 \right], \tag{3}$$

where $(i, j)$ denotes spatial position and $c \in \{1, \ldots, C\}$ indexes feature channels. The resulting energy map $E \in \mathbb{R}^{H/2 \times W/2}$ is bilinearly upsampled and min-max normalized per sample to obtain the final saliency map $A_{\text{wavelet}} \in [0, 1]^{H \times W}$.

This map serves as a proxy for local structural richness, highlighting regions of the latent space associated with high-frequency content (e.g., textures, contours, transitions). Unlike learned attention mechanisms based on semantic similarity (e.g., DINO (Caron et al., 2021)), our approach is deterministic and directly derived from signal properties. While we refer to $A_{\text{wavelet}}$ as an "attention map" for interpretability, it is best understood as a frequency-aware saliency measure.

Notably, our VAE fine-tuning objective (Eq. 1) helps suppress high-frequency artifacts and stabilize spectral behavior. This preprocessing step ensures that high-frequency activations captured by the DWT correspond to meaningful structure, rather than encoding noise, thereby improving the utility of $A_{\text{wavelet}}$ for guiding spatial supervision.

### 3.3 Adaptive Flow Matching with Frequency-Guided Masking

We adopt a continuous-time flow-matching formulation (Lipman et al., 2023; Esser et al., 2024) for training the latent diffusion model. Given a target latent $z_0$ and noise sample $\epsilon \sim \mathcal{N}(0, I)$, we define interpolated samples as:

$$z_t = (1 - t)\, z_0 + t\, \epsilon, \quad t \in [0, 1], \tag{4}$$

and supervise the predicted velocity field $v_\Theta(z_t, t, y)$, conditioned on text $y$, via:

$$\mathcal{L}_{\text{fm}} = \|(\epsilon - z_0) - v_\Theta(z_t, t, y)\|_2^2. \tag{5}$$

To incorporate frequency-awareness into the training objective, we apply spatially adaptive masking based on the wavelet saliency map $A_{\text{wavelet}}$. Specifically, for each location $(i, j)$, we define a time-dependent binary mask:

$$M_t(i, j) = \begin{cases} 1 & \text{if } T \cdot (A_{\text{wavelet}}(i, j) + \ell) \geq t \\ 0 & \text{otherwise} \end{cases}, \tag{6}$$

where $T$ is the total number of diffusion timesteps and $\ell \in (0, 1)$ sets a lower bound on refinement. This ensures that all regions receive at least $\ell T$ steps of supervision, while high-frequency regions benefit from extended refinement.

While selective spatial supervision has been investigated using transformer attention (Moser et al., 2025), our wavelet-derived saliency offers a fundamentally different perspective based on signal processing principles rather than learned semantic features, offering computational advantages and a more generalizable solution.

The final masked loss becomes:

$$\mathcal{L}_{\text{masked}} = \|M_t \odot [(\epsilon - z_0) - v_\Theta(z_t, t, y)]\|_2^2, \tag{7}$$

where $\odot$ denotes element-wise multiplication. This formulation focuses learning capacity on detail-rich regions, improves fidelity in high-frequency content, and does so without increasing inference complexity. Crucially, this mechanism operates purely at the objective level and is compatible with any latent diffusion model using a flow-based or score-based trajectory.

## 4 Experiments

### 4.1 Experimental Setup

**Datasets.** We evaluate LWD on two ultra-resolution datasets. Aesthetic-4K Zhang et al. (2025) is a curated 4K benchmark with GPT-4o-generated captions and high visual quality. LAION-High-Res is a filtered subset of LAION-5B Schuhmann et al. (2022), from which we sample 50K 2K-resolution and 20K 4K-resolution image-caption pairs. These two datasets differ in both visual and linguistic distributions, allowing us to assess both generation fidelity and generalization under caption variance.

**Implementation Details.** We implement LWD using PyTorch and `pytorch-wavelets` (Cotter, 2019) for the Haar-based DWT. For the masking strategy, we set the lower bound $\ell = 0.3$, selected via ablation to ensure each spatial location receives at least 30% of supervision.

**Evaluation Protocol.** To evaluate LWD as a holistic framework, all 'LWD + Model' variants utilize the Scale-Consistent VAE fine-tuning described in Section 3.1, while baseline models are evaluated using their original, off-the-shelf VAE checkpoints. Notably, we observed that LWD significantly accelerates convergence; models required only 10–50% of the training iterations suggested in their original papers to reach convergence. Detailed hyperparameters, training costs, and other configurations are provided in Appendix D.

**Evaluation Metrics.** We evaluate ultra-resolution text-to-image generation across three key dimensions. For image quality, we use Fréchet Inception Distance (FID) and LPIPS (lower is better), alongside the Gray Level Co-occurrence Matrix (GLCM) Score for texture and JPEG Compression

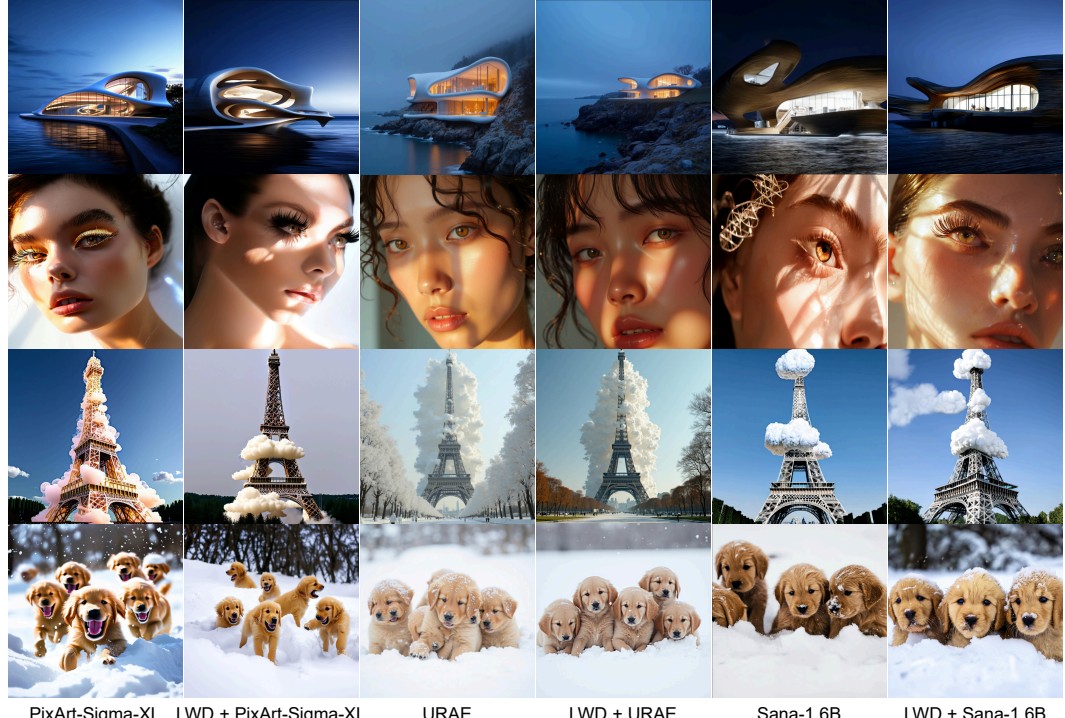

PixArt-Sigma-XL    LWD + PixArt-Sigma-XL    URAE    LWD + URAE    Sana-1.6B    LWD + Sana-1.6B

Figure 4: Visual comparison of 2K image generations. LWD demonstrates improved detail preservation in complex areas while avoiding over-sharpening or texture collapse.

Table 1: Quantitative results on different metrics. The prompts are from the HPD (Wu et al., 2023) dataset. All images are at a resolution of $2048 \times 2048$.

| Model | FID ↓ | LPIPS ↓ | MAN-IQA ↑ | QualiCLIP ↑ | HPSv2.1 ↑ | PickScore ↑ |
|---|---|---|---|---|---|---|
| SDEdit Meng et al. (2022) | 35.59 | 0.6456 | 0.3736 | 0.4480 | 30.92 | 22.86 |
| I-Max Du et al. (2024b) | 36.28 | 0.6750 | 0.3641 | 0.4139 | 30.62 | **23.02** |
| Diffusion-4K Zhang et al. (2025) | 37.10 | 0.6920 | 0.3550 | 0.4815 | 30.55 | 22.80 |
| PixArt-Sigma-XL Chen et al. (2024) | 36.58 | 0.6801 | 0.2949 | 0.4438 | 30.66 | 22.92 |
| Sana-1.6B Xie et al. (2025) | 35.75 | 0.7169 | 0.3666 | 0.5796 | 30.42 | 22.83 |
| Lumina-Image 2.0 Qin et al. (2025) | 54.96 | 0.6445 | 0.3663 | 0.4567 | 23.08 | 21.15 |
| FLUX-1.dev Labs (2024) | 37.58 | 0.6371 | **0.4110** | **0.5468** | 28.73 | 22.68 |
| URAE Yu et al. (2025) | 35.25 | 0.6717 | 0.4076 | 0.5423 | **31.15** | 22.41 |
| LWD + URAE | **32.88** | **0.6336** | 0.4099 | 0.5356 | 28.78 | 22.43 |

Ratio as a proxy for fine-grained detail. For semantic alignment, we report CLIPScore (Hessel et al., 2021) and QualiCLIP Agnolucci et al. (2024). Finally, for perceptual quality, we use MAN-IQA Yang et al. (2022), HPSv2.1 Wu et al. (2023), and PickScore Kirstain et al. (2023). Higher values indicate better performance for all metrics except FID and LPIPS.

## 4.2 QUANTITATIVE RESULTS

**2K Results.** Table 1 and the top block of Table 2 demonstrate consistent improvements from integrating LWD across multiple backbone models. On the HPD prompt dataset (Wu et al., 2023), LWD reduces FID by up to 7% and LPIPS up to 6% while also achieving comparable MAN-IQA and QualiCLIP, indicating improved semantic alignment and perceptual quality. Moreover, on the Aesthetic dataset (Zhang et al., 2025), these gains are observed across diverse architectures, reinforcing the generality and model-agnostic nature of our approach.

**4K Results.** On the Aesthetic-4K (the bottom block of Table 2) and HPD prompt dataset (Table 7), LWD outperforms baselines such as URAE and PixArt-Sigma, particularly in metrics like FID, CLIPScore and Aesthetics. The improvements are especially pronounced in regions with fine structural detail, such as hair, foliage, or architectural elements, highlighting LWD's ability to scale effectively to ultra-high resolutions. These results suggest that frequency-aware supervision provides

Table 2: Quantitative benchmarks of latent diffusion models on Aesthetic-Eval at $2048 \times 2048$ and $4096 \times 4096$. GLCM Score measures high-frequency texture richness using gray-level co-occurrence matrices, while Compression Ratio assesses visual complexity via JPEG file size heuristics, both introduced in Diffusion-4K Zhang et al. (2025).

|  | Model | FID ↓ | CLIPScore ↑ | Aesthetics ↑ | GLCM Score ↑ | Compression Ratio ↓ |
|---|---|---|---|---|---|---|
| **2K** | SD3-F16 Esser et al. (2024) | 43.82 | 31.50 | 5.91 | 0.75 | **11.23** |
|  | SD3-Diff4k-F16 Zhang et al. (2025) | 40.18 | 34.04 | 5.96 | **0.79** | 11.73 |
|  | LWD + SD3-F16 | **38.74** | **34.94** | **6.17** | 0.74 | 11.99 |
|  | PixArt-Sigma-XL Chen et al. (2024) | 39.13 | 35.02 | 6.43 | 0.79 | 13.66 |
|  | LWD + PixArt-Sigma-XL | **36.14** | 35.21 | 6.27 | **0.87** | **6.05** |
|  | Sana-1.6B Xie et al. (2025) | **32.06** | 35.28 | 6.15 | **0.93** | 24.01 |
|  | LWD + Sana-1.6B | 34.30 | **35.58** | **6.23** | 0.78 | 27.34 |
| **4K** | SD3-F16 Esser et al. (2024) | - | 33.12 | 5.97 | 0.73 | 11.97 |
|  | SD3-Diff4k-F16 Zhang et al. (2025) | - | 33.41 | 5.97 | 0.70 | **11.90** |
|  | LWD + SD3-F16 | - | **34.08** | **6.03** | **0.77** | 12.27 |
|  | Sana-1.6B Xie et al. (2025) | - | 34.40 | 6.14 | 0.39 | 48.36 |
|  | LWD + Sana-1.6B | - | **34.59** | **6.21** | **0.60** | **32.62** |

meaningful guidance even in challenging high-frequency regimes where baseline methods often struggle. LWD also achieves the highest GLCM score when paired with SD3-F16, and substantially improves Sana's performance on both GLCM and compression ratio, indicating stronger fine-detail retention and texture fidelity, without compromising overall quality.

## 4.3 QUALITATIVE RESULTS

Figures 4, 5, and 9 compare outputs from LWD and baseline models under identical prompts. LWD consistently renders sharper textures in high-frequency regions, such as fabric, skin, and hair, while preserving global structure.

Zoomed-in comparisons highlight improved reconstruction of fine details (e.g., hair strands, eyelashes, small objects) that are often blurred or omitted by baselines. These results suggest that frequency-aware masking not only enhances local precision but does so in a context-sensitive manner, avoiding over-sharpening or artifacts. This indicates that LWD effectively balances fine detail refinement with structural coherence. More full-resolution results can be found in Appendix B.

## 4.4 ABLATION STUDIES

### 4.4.1 EFFECT OF SCALE-CONSISTENCY LOSS ON RECONSTRUCTION QUALITY

Table 3 reports quantitative reconstruction metrics for various VAEs, with and without the proposed Scale-Consistency (SC) loss, evaluated on the Aesthetic-4K validation set. Across different architectures SD3-VAE, Flux-VAE, and Sana-AE, the addition of SC consistently improves performance, particularly in rFID and perceptual quality (LPIPS). For instance, Flux-VAE-SC outperforms its baseline with a significant reduction in rFID (0.50 vs. 0.73) and an increase in PSNR and SSIM, indicating sharper and more faithful reconstructions. Notably, SD3-VAE-F16-SC achieves a substantial LPIPS improvement (0.18 vs. 0.30), suggesting better perceptual fidelity despite using a more aggressive compression factor (F16). These results confirm that scale-consistent regularization enhances latent representations, making them more robust and structurally aligned, critical properties for downstream diffusion tasks.

### 4.4.2 IMPACT OF VAE REGULARIZATION AND WAVELET MASKING

To rigorously evaluate the contribution of each component within the LWD framework, we conducted a detailed ablation study. Our method comprises two main stages: (1) fine-tuning a VAE with a scale-consistency (SC) loss and (2) fine-tuning the diffusion model with our proposed wavelet-masked loss ($\mathcal{L}_{\text{masked}}$). The study presented here isolates the impact of each component on the final text-to-image generation quality.

The results, shown in Table 4, confirm that both the SC loss and the wavelet-masked loss contribute meaningfully to generation quality, with the full LWD framework yielding the strongest performance.

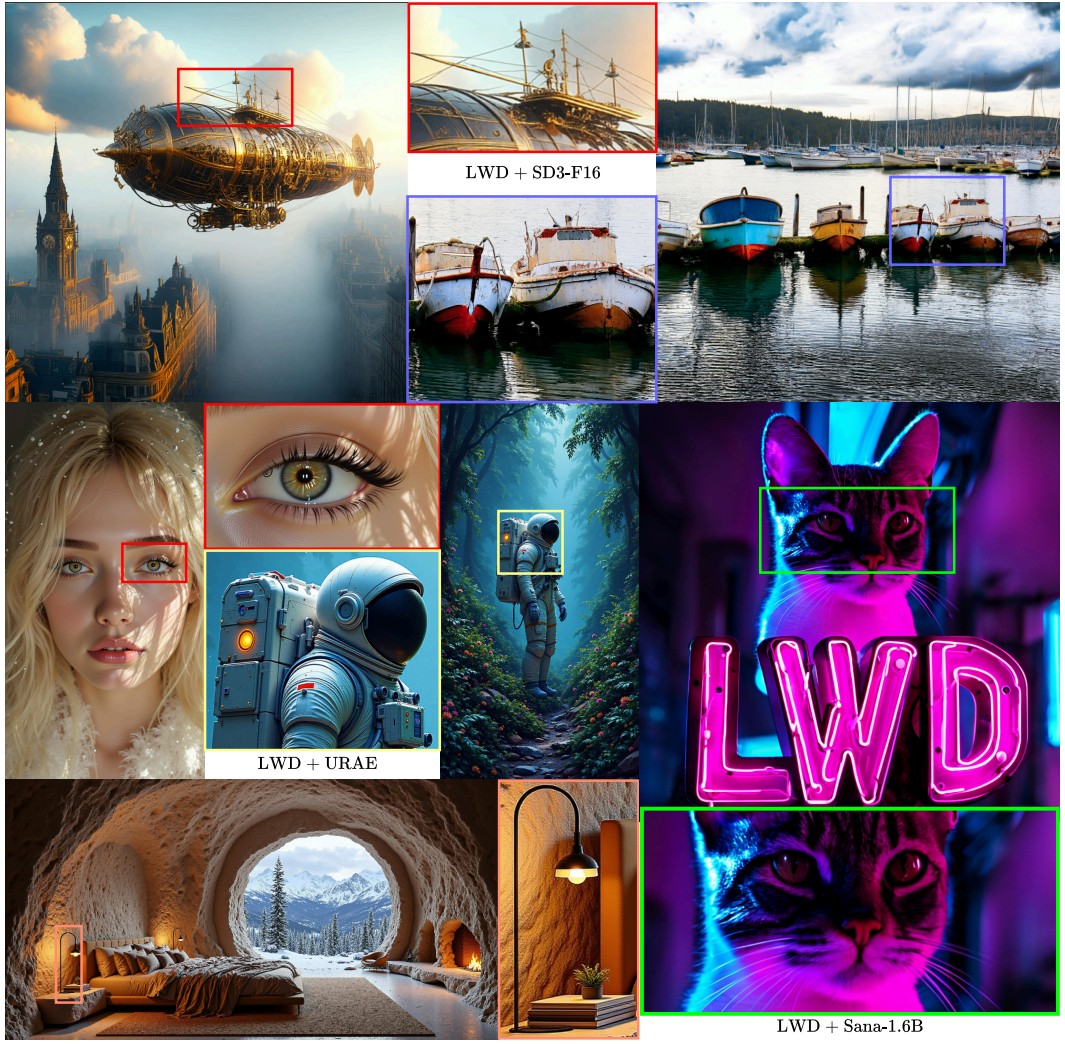

Figure 5: 4K images generated by LWD with different architectures.

Table 3: Quantitative reconstruction results of VAEs with and without the Scale-Consistency Loss 3.1 on Aesthetic-4K (Zhang et al., 2025) validation set.

| Model | rFID ↓ | NMSE ↓ | PSNR ↑ | SSIM ↑ | LPIPS ↓ |
|---|---|---|---|---|---|
| SD3-VAE (Esser et al., 2024) | 1.05 | 0.01 | **26.54** | **0.86** | **0.08** |
| SD3-VAE-F16 (Zhang et al., 2025) | 0.70 | 0.07 | 19.82 | 0.63 | 0.30 |
| SD3-VAE-F16-SC | 0.70 | **0.04** | 22.58 | 0.75 | 0.18 |
| Flux-VAE (Labs, 2024) | 0.73 | 0.01 | 27.18 | 0.89 | 0.07 |
| Flux-VAE-SC | **0.50** | 0.01 | **28.14** | **0.90** | **0.06** |
| Sana-AE (Xie et al., 2025) | 0.74 | 0.04 | 22.16 | 0.70 | **0.159** |
| Sana-AE-SC | **0.55** | **0.02** | **23.64** | **0.73** | 0.163 |

The results in Table 4 show that the GLCM score slightly decreases with the full LWD framework. This reflects a known limitation of classical texture metrics like GLCM, which do not always correlate with perceptual coherence. Our method trades raw statistical complexity for more realistic details, a positive trade-off validated by significant improvements in perceptual metrics like FID and Aesthetics.

Table 4: Ablation Study on the Contributions of LWD Components. We evaluate each component's impact on final generation quality using the Diffusion4k backbone on the Aesthetic dataset at 2048×2048 resolution.

| Configuration | FID ↓ | CLIPScore ↑ | Aesthetics ↑ | GLCM Score ↑ |
|---|---|---|---|---|
| Baseline (SD3-Diff4k-F16) | 40.18 | 34.04 | 5.96 | **0.79** |
| + VAE Scale-Consistency | 39.50 | 34.10 | 6.05 | 0.78 |
| + Wavelet Masking | 39.20 | 34.50 | 6.10 | 0.75 |
| **Full LWD (Ours)** | **38.74** | **34.94** | **6.17** | 0.74 |

## 4.5 DISCUSSION

Across quantitative and qualitative benchmarks, LWD enhances ultra-resolution image synthesis by integrating signal-derived saliency into the training loss. Compared to both conventional models and prior wavelet-based methods (e.g., Diffusion-4K), it improves perceptual fidelity, semantic alignment, and spectral regularity, without increasing inference cost or modifying the underlying architecture. Its plug-and-play nature makes it broadly compatible with modern latent diffusion pipelines.

LWD improves perceptual fidelity while maintaining comparable alignment scores. This reflects an intentional design choice: LWD prioritizes high-frequency detail recovery to prevent texture collapse at UHR scales, complementing the base model's semantic capabilities rather than replacing them.

Beyond quality gains, LWD represents a shift toward more interpretable and structure-aware supervision. Unlike attention mechanisms that rely on semantic priors, LWD leverages wavelet energy as a transparent, self-supervised signal to prioritize detail-rich regions.

This frequency-guided supervision introduces a form of spatial curriculum learning, where complex regions receive more focused updates. Such adaptive loss weighting opens avenues for dynamic training strategies, such as frequency-aware learning rates or hybrid spatial-frequency schedules. These mechanisms may be especially valuable in domains where structural detail is critical but semantic guidance is weak, such as scientific visualization, material design, or multimodal generation.

**Limitations and Future Work.** While LWD improves generation quality without architectural changes or inference overhead, it inherits limitations common to latent diffusion models. In particular, VAE compression can cause the loss of fine-grained semantic detail, potentially limiting performance in tasks requiring precise spatial alignment or photorealistic accuracy.

Future work could address this by incorporating higher-fidelity latent spaces or hybrid approaches that combine latent and pixel-space supervision. Extending LWD to domains such as video generation, depth-aware synthesis, or multimodal conditioning also represents a promising direction. The frequency-aware masking mechanism is general and could be adapted to guide temporal attention, cross-modal alignment, or resolution-specific sampling in broader generative contexts.

## 5 CONCLUSION

We introduced *Latent Wavelet Diffusion* (LWD), a general and modular framework that integrates frequency-based supervision into latent diffusion models for ultra-high-resolution image synthesis. By computing wavelet energy maps in the latent space and applying spatially and temporally adaptive masking, LWD selectively emphasizes high-detail regions during training. Without requiring architectural modifications or incurring additional inference cost, LWD consistently improves perceptual fidelity and semantic alignment across models such as Flux and SD3. It preserves high-frequency detail and structural coherence more effectively than prior methods, demonstrating the value of signal-aware supervision in guiding the generative process. By unifying principles from signal processing and diffusion modeling, LWD offers a scalable and interpretable approach applicable to a wide range of generative architectures.

**Broader Impact.** LWD promotes efficient and interpretable generation by aligning supervision with signal-level detail. This may benefit applications requiring controllable high-resolution synthesis, while raising familiar concerns around synthetic media misuse. Incorporating safeguards and provenance tools remains an important direction.

## REPRODUCIBILITY STATEMENT

To ensure the reproducibility of our work, we have included the core Python script detailing our wavelet-based masking algorithm in the supplementary material. Further implementation details, including key hyperparameters and computational requirements, are provided in Appendix section D. We release our public GitHub repository[1], which contains the complete implementation, training scripts, evaluation code, and the final pre-trained model checkpoints.

## ETHICAL STATEMENT

Our work is built upon publicly available datasets commonly used in the field of generative modeling. We acknowledge that these large-scale datasets may contain inherent societal biases, which our model could potentially learn and reproduce. We have used these datasets in accordance with their original licenses. Below, we discuss the potential societal impacts of our work.

### POSITIVE IMPACT

The primary goal of our research is to advance the state of high-resolution image generation, which has significant positive applications. These include empowering artists, designers, and content creators with more powerful creative tools; enhancing visual effects for entertainment and media; and potentially aiding in scientific visualization and data augmentation. By developing a method that improves quality with **zero inference overhead**, we aim to make high-fidelity generative AI more accessible and practical for a wider range of beneficial uses.

### POTENTIAL RISKS AND MITIGATION

We recognize that generative models can be misused for malicious purposes, such as creating misinformation ("deepfakes"), generating harmful or explicit content, and perpetuating societal biases. To mitigate these risks, we are committed to the following measures:

1. **Responsible Release:** We release our code and models under a responsible AI license (e.g., a variant of the CreativeML Open RAIL-M license) that explicitly prohibits use for malicious, harmful, or unethical purposes.
2. **Acknowledging Limitations:** We are transparent about the limitations of our model and its potential to generate biased or factually incorrect content, as discussed in the main paper.
3. **Encouraging Safe Deployment:** We strongly encourage all downstream users of our models to implement their own safety filters, content moderation systems, and ethical guidelines before deploying any applications.

## LLM USAGE STATEMENT

A large language model (LLM) was used as a writing assistance tool during the preparation of this manuscript. The LLM's role was limited to improving grammar, clarity, and conciseness. All content was conceived and written by the authors, who take full responsibility for the paper's scientific integrity.

## ACKNOWLEDGEMENT

The work of L. Sigillo was partially supported by "Ricerca e innovazione nel Lazio - incentivi per i dottorati di innovazione per le imprese e per la PA - L.R. 13/2008" of Regione Lazio, Project "Deep Learning Generativo nel Dominio Ipercomplesso per Applicazioni di Intelligenza Artificiale ad Alta Efficienza Energetica", under grant number 21027NP000000136. The work of D. Comminiello was partly supported by Progetti di Ateneo of Sapienza University of Rome under grant numbers RM123188F75F8072 and RM1241910FC4BEEA and by the European Union under the Italian

---

[1]https://github.com/LuigiSigillo/LatentWaveletDiffusion

National Recovery and Resilience Plan (NRRP) of NextGenerationEU, "Rome Technopole" (CUP B83C22002820006)—Flagship Project 5: "Digital Transition Through AESA Radar Technology, Quantum Cryptography and Quantum Communications".

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

# A   THEORETICAL AND IMPLEMENTATION DETAILS OF WAVELET-GUIDED MASKING

This section provides a detailed theoretical background on the Discrete Wavelet Transform (DWT) and its application in our framework, discusses specific implementation choices and hyperparameters, and analyzes the properties and trade-offs of our proposed method.

## A.1   THEORETICAL FOUNDATION: DWT AND RELEVANCE MAPS

**Discrete Wavelet Transform (DWT).**   The Discrete Wavelet Transform (DWT) decomposes a 2D signal, such as an image or a latent tensor, into orthogonal frequency subbands by applying separable cascades of low-pass and high-pass filters along both spatial dimensions. Given an input $\mathbf{X} \in \mathbb{R}^{H \times W \times D}$, a single-level DWT produces four spatial subbands:

- **LL** (Low-Low): Approximation coefficients capturing global structure and coarse semantics.
- **LH** (Low-High): Horizontal detail coefficients, sensitive to vertical edges.
- **HL** (High-Low): Vertical detail coefficients, highlighting horizontal edges.
- **HH** (High-High): Diagonal detail coefficients, encoding fine textures and high-frequency transitions.

Formally, the DWT of $\mathbf{X}$ is given by:

$$\text{DWT}(\mathbf{X}) = \begin{cases} \text{LL} = \mathbf{X} * h_{\text{low}} \downarrow 2 * h_{\text{low}} \downarrow 2 \\ \text{LH} = \mathbf{X} * h_{\text{low}} \downarrow 2 * h_{\text{high}} \downarrow 2 \\ \text{HL} = \mathbf{X} * h_{\text{high}} \downarrow 2 * h_{\text{low}} \downarrow 2 \\ \text{HH} = \mathbf{X} * h_{\text{high}} \downarrow 2 * h_{\text{high}} \downarrow 2 \end{cases}$$

where $h_{\text{low}}$, $h_{\text{high}}$ are orthogonal wavelet filters (e.g., Haar or Daubechies), $*$ denotes convolution, and $\downarrow 2$ indicates downsampling by a factor of 2.

**Relevance Map Construction.**   To identify spatial regions that require enhanced refinement during generation, we compute a wavelet-based relevance map from the high-frequency subbands. Specifically, we calculate the aggregated energy of the directional detail coefficients:

$$\mathbf{M}_{\text{relevance}} = \text{LH}^2 + \text{HL}^2 + \text{HH}^2$$

This yields a saliency map that highlights localized frequency-rich regions, such as edges, textures, and fine details. The relevance map is then resized to match the original latent resolution via bilinear interpolation and normalized to the range [0, 1]:

$$\mathbf{M}_{\text{norm}} = \frac{\mathbf{M}_{\text{relevance}} - \min(\mathbf{M}_{\text{relevance}})}{\max(\mathbf{M}_{\text{relevance}}) - \min(\mathbf{M}_{\text{relevance}}) + \epsilon}$$

where $\epsilon$ is a small constant to avoid division by zero. The LL subband, which encodes coarse spatial content, offers limited information about local complexity and is thus excluded. In contrast, the aggregated energy of the LH, HL, and HH bands approximates the local gradient magnitude, similar in spirit to edge detectors like Sobel and Laplacian filters, and aligns with the intuition that visually salient regions often correspond to areas with rich high-frequency content. This approach is theoretically supported by prior work in multiscale signal analysis, which demonstrates that the local maxima of wavelet detail coefficients correspond to structural singularities and perceptual boundaries (Mallat & Hwang, 1992).

## A.2   IMPLEMENTATION AND HYPERPARAMETERS

**Choice of Wavelet Basis.**   We deliberately selected the Haar wavelet for its ideal trade-off of properties for our framework:

- **Computational Efficiency:** It is the most computationally efficient wavelet, minimizing training overhead.

- **Preservation of Discontinuities:** Its discontinuous nature makes it exceptionally effective at localizing and preserving sharp edges and contours.
- **Sparsity and Non-Redundancy:** Its orthogonality and compact support induce sparse, non-redundant representations, making our energy maps precise and ideal for our masking strategy.

**Comparative Analysis with Other Transforms.**   To empirically validate the selection of Haar wavelets over other frequency analysis methods, we conducted a rigorous ablation study comparing our approach against Daubechies wavelets (db2) and FFT-based High-Pass filtering. The results are summarized in Table 5.

Two key principles dictate the superior performance of Haar in this context:

1. **Spatial Localization:** LWD requires a spatially precise mask $M_t(i, j)$ to target specific latent regions. While global transforms like FFT/DCT provide excellent frequency resolution, they sacrifice spatial localization; a high-frequency coefficient corresponds to periodic patterns across the entire image, not specific positions. Recovering spatial energy maps via inverse transformation introduces Gibbs ringing near sharp transitions. These artifacts cause signal leakage into neighboring latent positions, blurring the mask and degrading texture precision (GLCM 0.71 vs 0.74).

2. **Compact Support:** Among wavelets, Haar has the most compact support (2 coefficients), minimizing cross-position interference. This is critical for generating sharp binary training masks. Smoother wavelets (e.g., Daubechies) introduce wider receptive fields, creating "gray areas" at mask boundaries that dilute supervision without semantic benefit.

As shown in Table 5, while FFT is computationally faster due to hardware optimizations, the spatial artifacts degrade generation quality (FID 39.45). Haar achieves the optimal balance, outperforming Daubechies in both efficiency and final texture fidelity.

Table 5: Ablation of Frequency Decomposition Methods. Comparison on Diffusion4k backbone (Aesthetic dataset, $2048 \times 2048$).

| Method | Cost (ms) $\downarrow$ | FID $\downarrow$ | Aesthetics $\uparrow$ | GLCM $\uparrow$ |
|---|---|---|---|---|
| LWD (Haar) | 1.136 | **38.74** | **6.17** | **0.74** |
| LWD (Daubechies) | 1.274 | 38.92 | 6.14 | 0.73 |
| LWD (FFT High-Pass) | **0.875** | 39.45 | 6.08 | 0.71 |

**Wavelet Masking Lower Bound.**   The primary hyperparameter for our wavelet masking strategy is the lower bound $l$. The value $l = 0.3$ was chosen based on an ablation study (Table 6). This study revealed a clear trade-off: a very low value (e.g., $l < 0.1$) can cause smooth regions to be under-trained, while a very high value (e.g., $l > 0.7$) diminishes the benefit of targeted refinement, causing performance to regress towards the baseline. The value $l = 0.3$ was found to be a robust sweet spot.

Table 6: Ablation on the Masking Lower Bound ($l$).

| Masking Lower Bound ($l$) | FID $\downarrow$ | GLCM Score $\uparrow$ | CLIPScore $\uparrow$ |
|---|---|---|---|
| 0.0 | 34.15 | 0.68 | 0.5411 |
| 0.1 | 33.21 | 0.72 | 0.5420 |
| **0.3** | **32.88** | **0.74** | **0.5423** |
| 0.5 | 33.46 | 0.71 | 0.5419 |
| 0.7 | 34.02 | 0.69 | 0.5407 |

**Intuition for the Masking Strategy.**   The time-dependent masking schedule is designed to allocate more training attention to structurally rich regions of the image, which are identified via higher

wavelet energy. The schedule ensures that these areas are refined over more training steps, while still providing a minimum level of supervision to all regions. This enhances high-frequency details without overfitting to them.

### A.3    ANALYSIS OF METHOD PROPERTIES AND TRADE-OFFS

**Preservation of Global Structure.**    Our wavelet-masked loss is designed to preserve global coherence. The masking computation targets only the high-frequency subbands (LH, HL, HH), which encode localized detail. The LL subband, which captures low-frequency, global structure, is not involved. This ensures that while local refinement is emphasized, the global scene layout and object structure remain intact.

**Robustness and Potential Artifacts.**    Our wavelet-based masking strategy is agnostic to the source of high-frequency information and has proven robust across diverse scenes without introducing noticeable artifacts. The VAE fine-tuning stage is key to this stability, as it regularizes the latent space to ensure that the high-frequency energy used for masking corresponds to meaningful content rather than spurious signals. While we have not observed failure cases in our benchmarks, investigating domain-specific behavior is an important direction for future work.

**On the Synergy of Frequency Suppression and Utilization.**    A natural question arises from the apparent tension between our use of a multi-scale VAE loss to suppress spurious high-frequency components, and our later use of high-frequency energy to guide the adaptive masking. These two strategies serve complementary and sequential roles. The VAE loss does not eliminate all high-frequency content; rather, it penalizes frequency components inconsistent across scales, which often correspond to noise or artifacts. This regularization aligns the latent spectral distribution more closely with that of clean, natural images.

Crucially, it is precisely this filtered latent space that makes our frequency-based attention meaningful. Once the latent tensor is regularized, the remaining high-frequency energy (extracted via DWT) is more strongly correlated with visually salient features like edges and textures. In other words, by denoising the latent representation, the VAE enhances the signal-to-noise ratio of our wavelet attention mechanism. This sequential design, first purifying the latent space, then exploiting its structured frequency characteristics, ensures our method combines signal-domain regularization and content-adaptive computation in a synergistic manner.

## B    ADDITIONAL RESULTS FOR 4K

To assess the efficacy of our proposed Latent Wavelet Diffusion (LWD), we conduct a detailed evaluation focusing on the challenging 4K resolution (4096×4096).

### B.1    QUANTITATIVE RESULTS

Table 7 presents the quantitative comparison on 4K image generation. We evaluate the generated images using several key metrics: MAN-IQA (Yang et al., 2022) and QualiCLIP (Agnolucci et al., 2024), which assess perceptual image quality and alignment with textual prompts, respectively. Additionally, we compute the GLCM (Gray-Level Co-occurrence Matrix) score as a measure of texture complexity and detail richness in the generated high-resolution outputs. Finally, we report the Compression Ratio, indicating the compressibility of the generated images, which can be indicative of redundancy or lack of fine details.

Our LWD-enhanced URAE demonstrates competitive performance across all evaluated metrics. Notably, it achieves the highest MAN-IQA score and GLCM score, suggesting superior perceptual quality and richer textural details compared to the baselines. Furthermore, our LWD + URAE achieves a favorable Compression Ratio (28.77), better than URAE and PixArt-Sigma-XL, suggesting a good balance between detail and redundancy. These quantitative results underscore the effectiveness of our LWD approach in enhancing the visual quality and detail of images generated at 4K resolution.

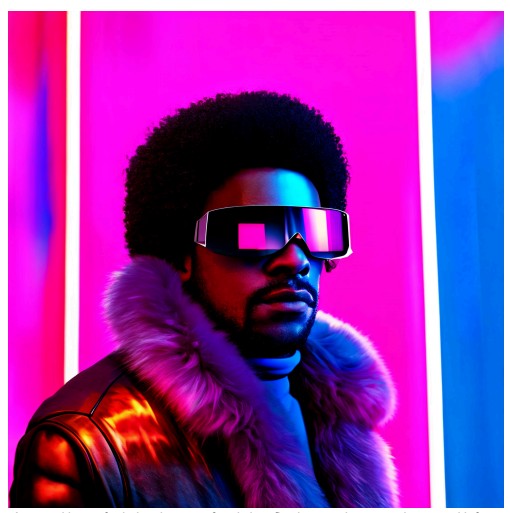

A man with an afro hairstyle wears futuristic reflective sunglasses and a coat with fur lining, standing in front of a vibrant pink and blue neon sign.

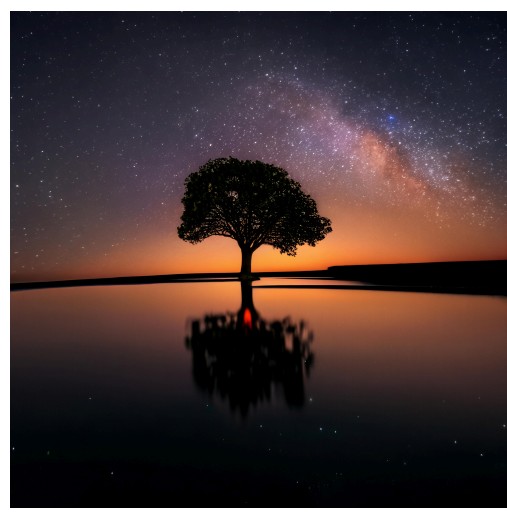

A lone tree stands in calm water reflecting the starry night sky, with the Milky Way stretching above and warm orange hues from a distant horizon.

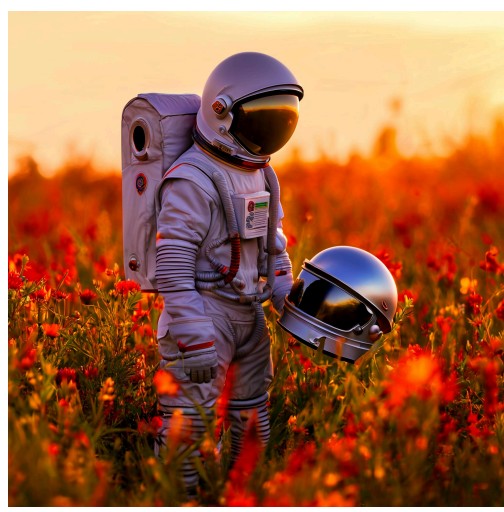

A young astronaut in a light-colored suit stands in a vibrant field of wildflowers, holding a helmet and gazing downward, with a warm, glowing sunset in the background.

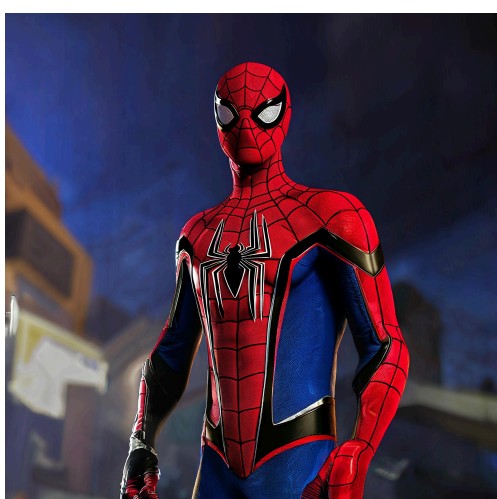

A person wearing a Spider-Man suit in the game Half-Life Alyx."

Figure 6: Images generated at 4K resolution with LWD+SANA.

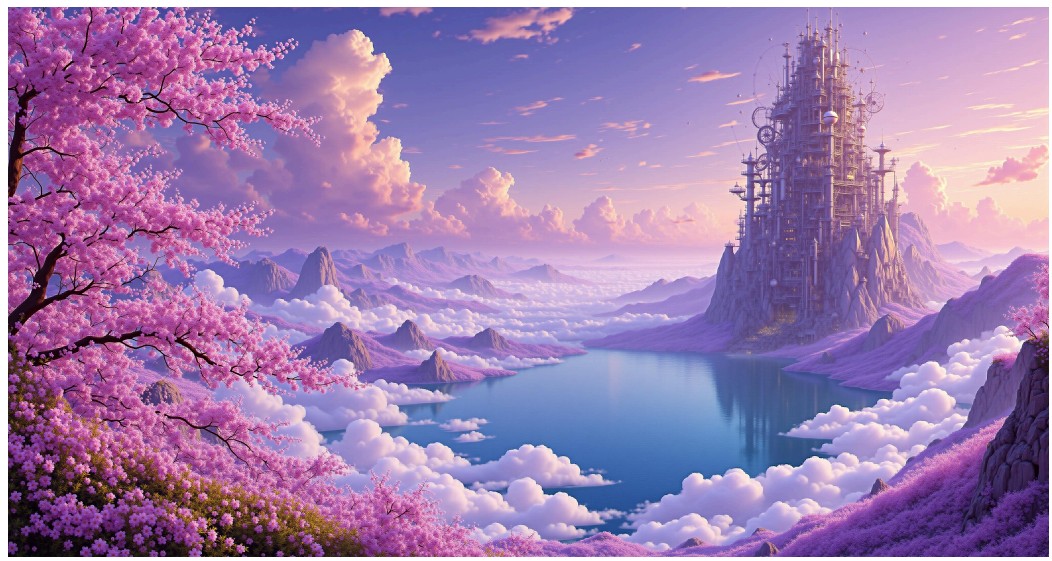

A surreal landscape depicting an ethereal fusion of natural beauty and fantastical architecture, reminiscent of Salvador Dali's dreamlike paintings. From above the clouds, one gazes upon a colossal tower emerging from the earth, its intricate gears visible as it merges seamlessly with a tranquil mountain lake. The scene is bathed in an otherworldly glow, casting lavender and gold hues across the sky, while delicate cherry blossoms flutter gently in the foreground, adding a sense of serenity to this breathtaking vision where time and nature intertwine.

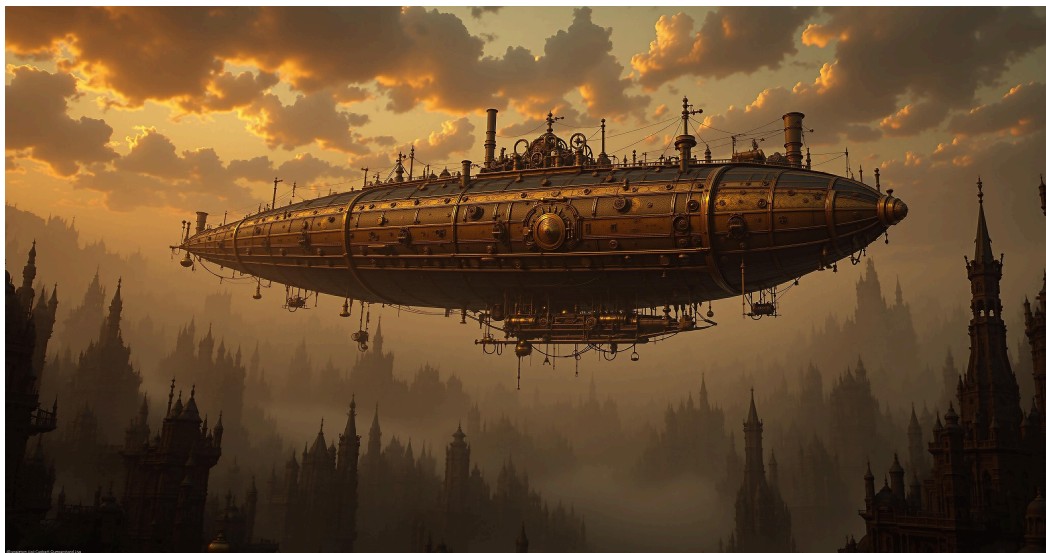

Steampunk airship floating above a misty Victorian cityscape, intricate brass and copper mechanical details, golden hour lighting, billowing clouds, detailed architectural elements, rich warm color palette, cinematic composition.

Figure 7: Images generated at 4K resolution with LWD+URAE.

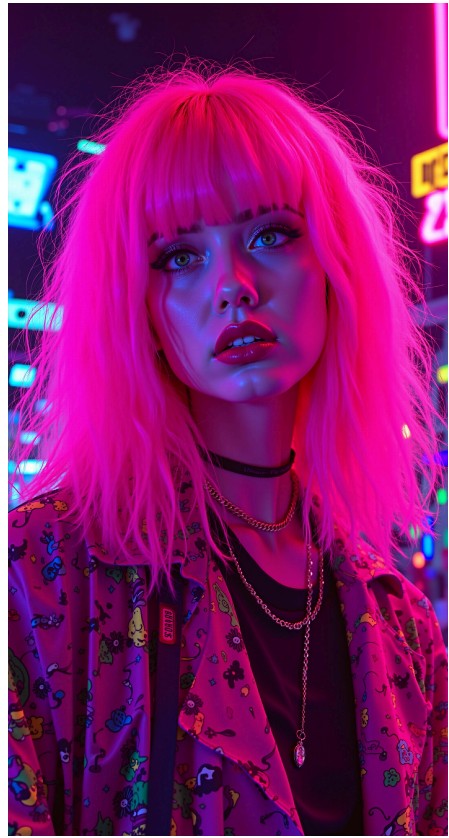

Girl with pink hair, vaporwave style, retro aesthetic, cyberpunk, vibrant, neon colors, vintage 80s and 90s style, highly detailed.

A sleek black luxury sedan parked on a rain-soaked city street at night, reflecting neon lights from nearby buildings. The wet pavement glistens, and the car's smooth curves are highlighted by the ambient glow of the urban environment.

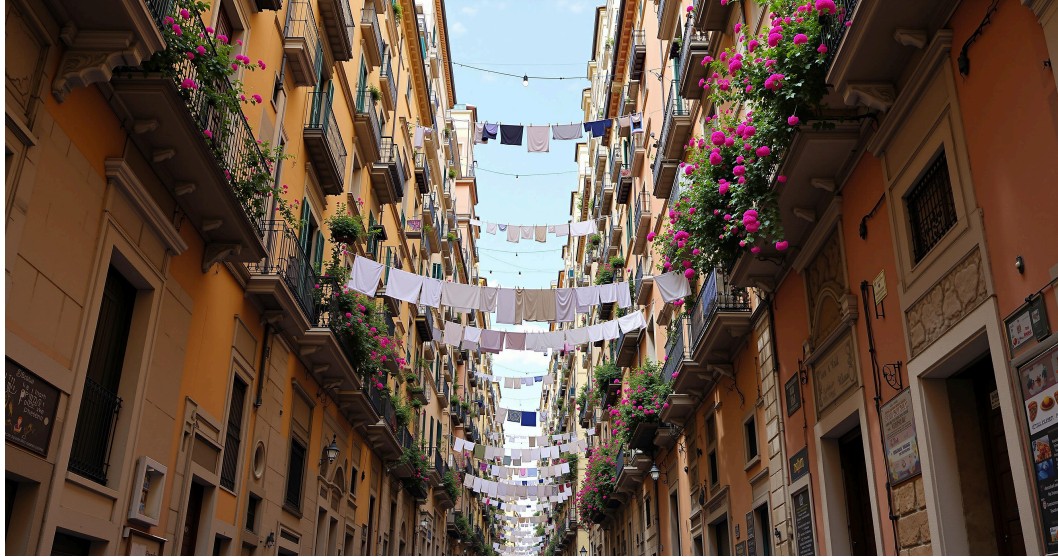

A narrow and picturesque alley in the historic center of Naples, with laundry hanging out to dry between flower-filled balconies and the inviting aroma of freshly baked pizza in the air.

Figure 8: Images generated at 4K resolution with LWD+URAE.

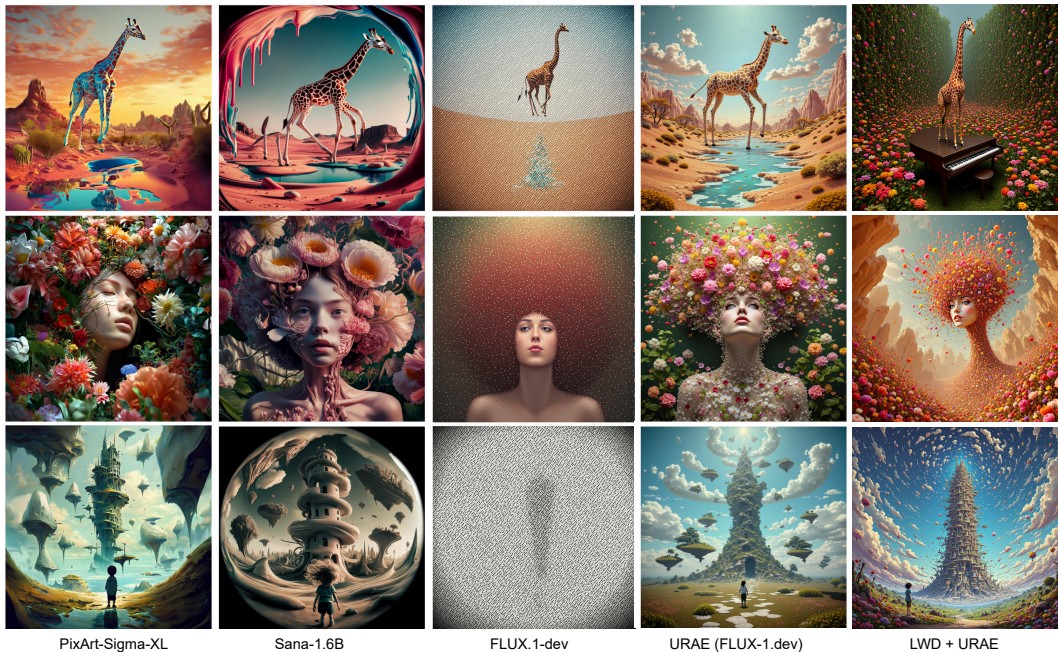

| PixArt-Sigma-XL | Sana-1.6B | FLUX.1-dev | URAE (FLUX-1.dev) | LWD + URAE |

Figure 9: Visual comparison of 4K image generations from LWD and competing baselines.

## B.2 QUALITATIVE ANALYSIS

To complement the quantitative evaluation, Figures 10, 11, 13, and 12 provide a qualitative comparison of LWD against selected baselines, presenting side-by-side comparisons of the generated images.

The visual comparisons highlight the benefits of our LWD enhancement. Our method demonstrates the generation of images with finer and more intricate details, particularly noticeable in complex textures and object boundaries. These qualitative observations align with our quantitative findings, reinforcing the effectiveness of the proposed LWD for high-resolution image generation.

Table 7: Evaluation on ultra resolution (4096 × 4096) image generation task with (Wu et al., 2023) prompts.

| Method | MAN-IQA (↑) | QualiCLIP (↑) | GLCM Score ↑ | Compression Ratio ↓ |
|---|---|---|---|---|
| PixArt-Sigma-XL (Chen et al., 2024) | 0.2935 | 0.2308 | 0.48 | 45.15 |
| Sana-1.6B (Xie et al., 2025) | 0.3288 | **0.4979** | 0.71 | **25.89** |
| FLUX-1.dev (Labs, 2024) | 0.3673 | 0.2564 | - | - |
| URAE (Yu et al., 2025) | 0.3850 | 0.3758 | 0.41 | 38.86 |
| LWD + URAE | **0.4011** | 0.3855 | **0.74** | 28.77 |

**Selection Protocol** All qualitative examples shown in the paper were generated following a strict, reproducible protocol to prevent cherry-picking. Prompts were sourced directly from the original papers of the baseline models (e.g., URAE, Diffusion-4K) or the HPD benchmark dataset. For each prompt, the displayed image is the first output generated using a fixed random seed, applied identically to both the baseline and our LWD-enhanced model. We have included our code in the supplementary materials to ensure full transparency. While subjective preferences for certain images may vary, our method consistently improves objective indicators of texture and detail.

## C FREQUENCY-AWARE EVALUATION

To rigorously assess the frequency characteristics of generated images, we propose a suite of frequency-sensitive metrics that extend beyond standard perceptual scores. These metrics are designed to quantify the degree to which generated images preserve the natural frequency distribution

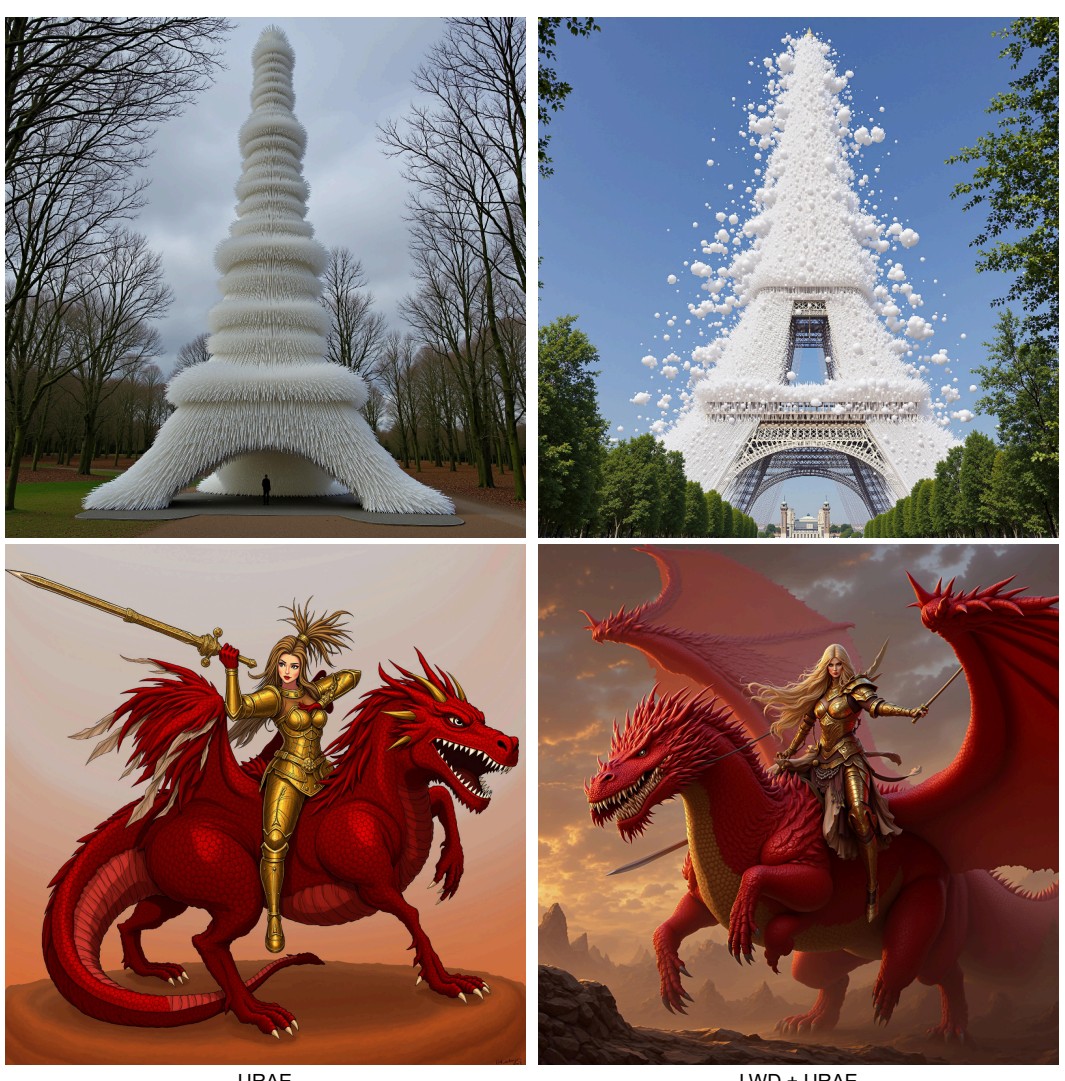

Figure 10: 4K generation of URAE vs LWD + URAE.
Upper caption: *"Eiffel Tower was Made up of more than 2 million translucent straws to look like a cloud, with the bell tower at the top of the building, Michel installed huge foam-making machines in the forest to blow huge amounts of unpredictable wet clouds in the building's classic architecture."*.
Lower caption: *"Barbarian woman riding a red dragon, holding a broadsword, in gold armour."*

observed in real images, with particular attention to the presence and quality of high-frequency details.

**High/Low Frequency Ratio (HLFR)**: We decompose each image using a 2D discrete wavelet transform (DWT) and compute the total energy in the detail coefficients (high-frequency subbands: LH, HL, HH) and in the approximation coefficients (low-frequency LL subband). The HLFR is defined as the ratio of high-frequency to low-frequency energy:

$$\text{HLFR} = \frac{E_{LH} + E_{HL} + E_{HH}}{E_{LL}}.$$

This ratio reflects the relative emphasis on fine-scale structures. A value similar to the reference (real images) indicates a natural distribution of frequency content. Large deviations can signal oversmoothing or hallucinated detail.

**Ratio Difference from Real (RDR)**: To quantify deviation from the natural HLFR, we compute the absolute difference between the HLFR of the generated image and the real reference:

$$\text{RDR} = \left|\text{HLFR}_{\text{gen}} - \text{HLFR}_{\text{real}}\right|.$$

Lower values are better, indicating better alignment with the natural frequency distribution.

**Wavelet Quality Score (WQS)**: This metric evaluates the structural similarity between generated and real images in the wavelet domain, where frequency components are explicitly separated by scale and orientation. Given a multi-level discrete wavelet transform (DWT) of both the generated image $I_g$ and the reference image $I_r$, we compute the SSIM and MSE for each corresponding subband $s$ across all decomposition levels $l$. The final WQS aggregates the per-subband scores using frequency-aware weights:

$$\text{WQS} = \sum_{l=1}^{L} \sum_{s \in \{LL, LH, HL, HH\}} w_{l,s} \cdot \text{SSIM}(I_r^{l,s}, I_g^{l,s}) - \lambda \cdot \text{MSE}(I_r^{l,s}, I_g^{l,s}),$$

where $w_{l,s}$ are weights that can prioritize perceptually important subbands (e.g., low-frequency LL or high-frequency HH), and $\lambda$ is a scaling factor that penalizes distortion. The score is normalized to $[0, 1]$, where 1 indicates perfect structural alignment. Higher WQS values reflect better reconstruction fidelity across frequency scales, meaning the model preserves both coarse structure and fine texture.

**High-Frequency Energy (HFE)**: This metric quantifies the total energy of the image's high-frequency components after wavelet decomposition. For a given decomposition level, we define:

$$\text{HFE} = \sum_{l=1}^{L} \left( \|I^{l,LH}\|^2 + \|I^{l,HL}\|^2 + \|I^{l,HH}\|^2 \right).$$

This value provides an absolute measure of fine-scale activity in the image. While real images have characteristic HFE ranges, excessive HFE may indicate artifacts or noise, and too little HFE suggests oversmoothing. Alignment with the real HFE is typically ideal.

**High-Frequency Emphasis Index (HFEI)**: This metric evaluates how much the model over- or under-emphasizes high-frequency content relative to the real distribution. We define it as:

$$\text{HFEI} = \left( \frac{\text{HFE}_{\text{gen}}}{\text{TotalEnergy}_{\text{gen}}} \right) - \left( \frac{\text{HFE}_{\text{real}}}{\text{TotalEnergy}_{\text{real}}} \right),$$

where total energy is computed over all wavelet subbands. HFEI $> 0$ indicates the generated image places more emphasis on high frequencies than real images (potentially hallucinated detail), while HFEI $< 0$ indicates a loss of fine detail. An HFEI near zero is ideal.

**Perceptual Metrics**: For completeness, we also report FSIM (Zhang et al., 2011) and MS-SSIM (Wang et al., 2003), which capture visual similarity and structural coherence, respectively. Both metrics are bounded between 0 and 1, with higher values indicating better perceptual quality.

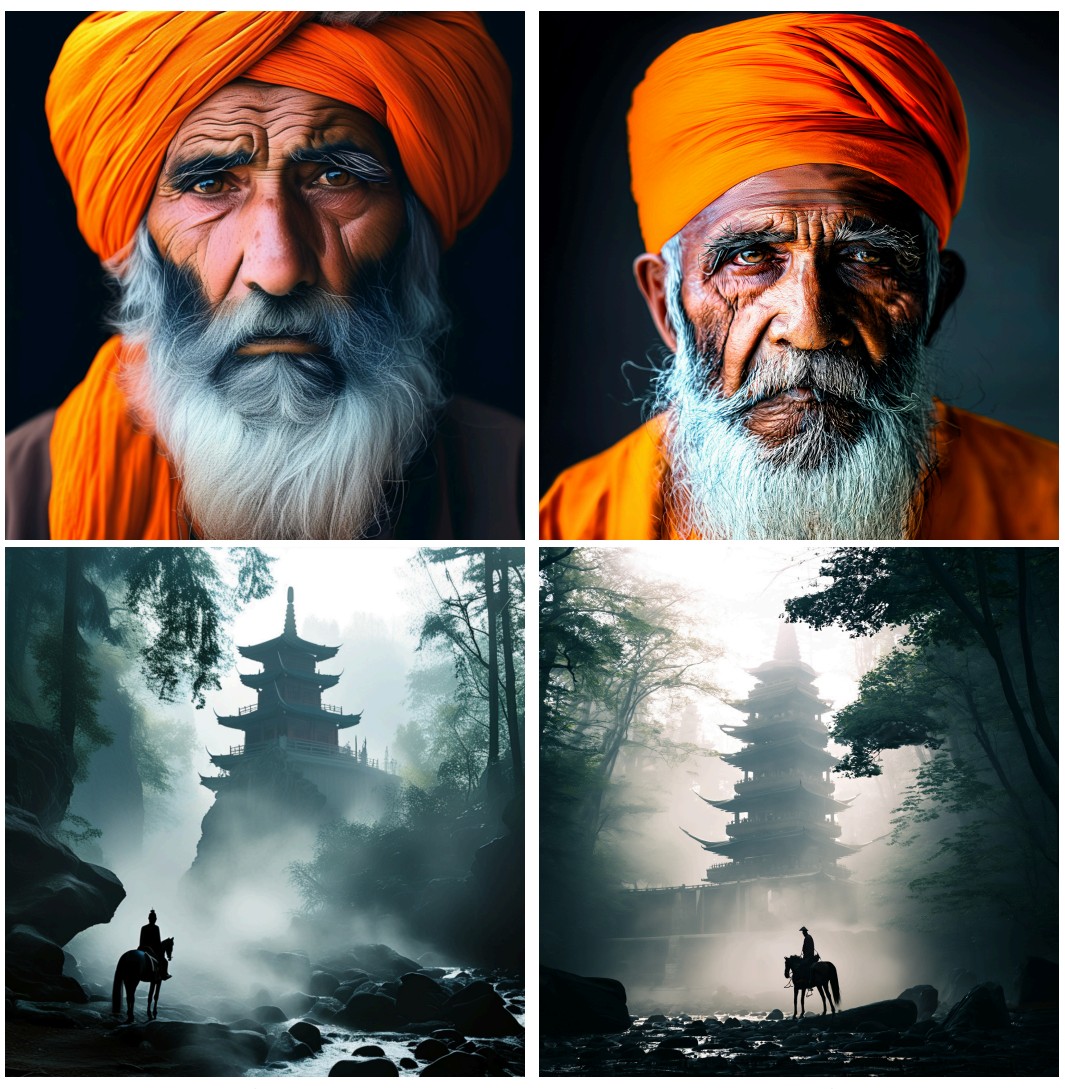

PixArt-Sigma-XL                    LWD + PixArt-Sigma-XL

Figure 11: 2K generation of PixArt-Sigma-XL vs LWD + PixArt-Sigma-XL.
Upper caption: *"An elderly man with a prominent, bushy beard and deep-set eyes wears a vibrant orange turban, his weathered face marked by lines of age and experience."*. Lower caption: *"A lone figure on a horse stands in a misty forest, gazing up at a tall, multi-tiered temple surrounded by towering trees and soft, diffused light. Steam rises from the rocks near a stream, creating an atmospheric scene of tranquility and mystery."*

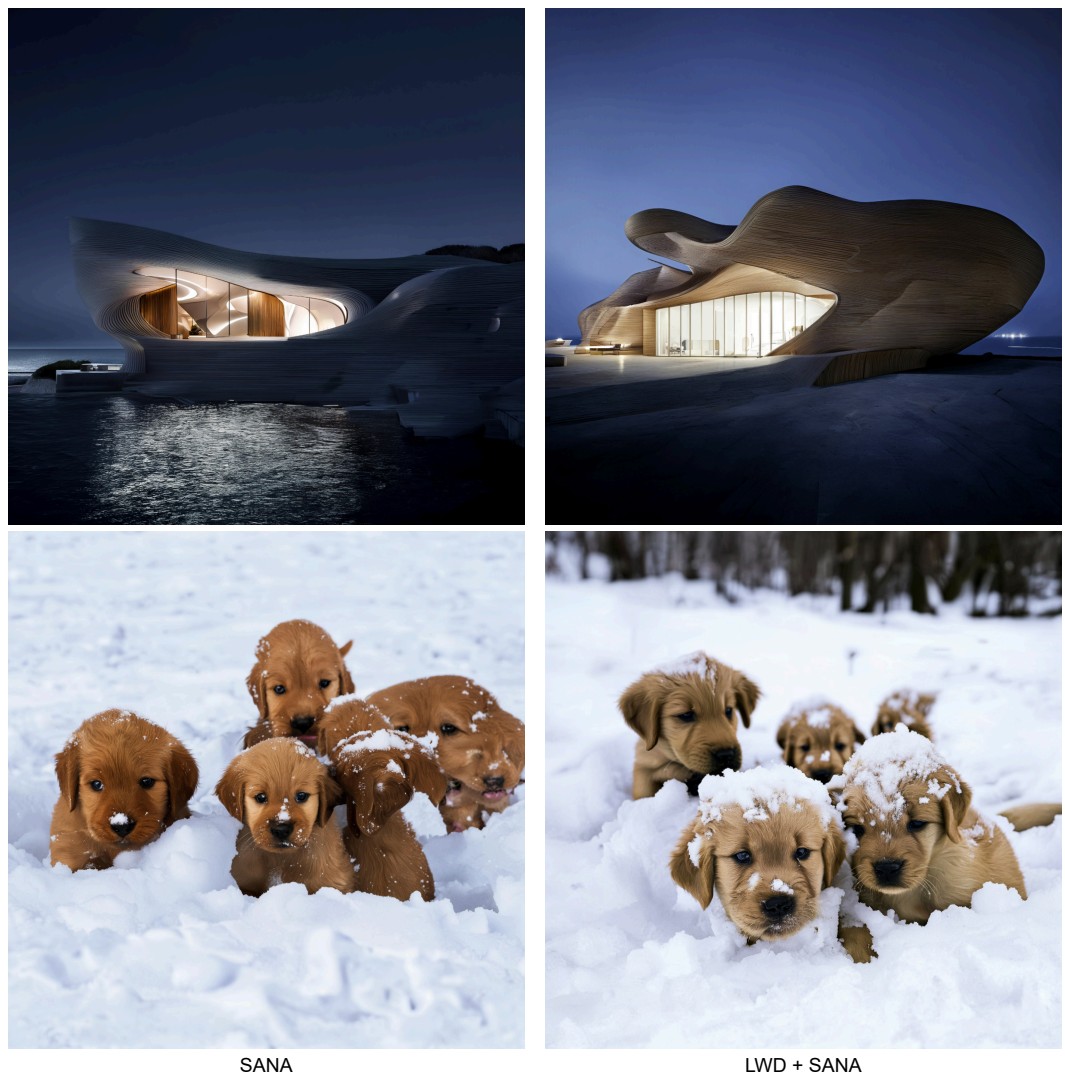

SANA                                    LWD + SANA

Figure 12: 4K generation of Sana vs LWD + Sana.
Upper caption: *"A litter of golden retriever puppies playing in the snow. Their heads pop out of the snow, covered in."*. Lower caption: *"A curvy timber house near a sea, designed by Zaha Hadid, represent the image of a cold, modern architecture, at night, white lighting, highly detailed."*

**Results:** Table 8 summarizes the performance of diffusion backbones, along with their respective versions enhanced with LWD. LWD consistently improves frequency alignment: for instance, the LWD+Diff4K variant achieves a high/low frequency ratio of 0.0556, nearly identical to the real reference (0.0560), while also minimizing the ratio difference (0.0438). Wavelet Quality Scores improve in both cases, and the HF energy is more tightly controlled, preventing oversharpening. Importantly, LWD preserves or improves FSIM and MS-SSIM, confirming that frequency fidelity does not come at the expense of perceptual quality. These results demonstrate that LWD enhances the frequency realism of generated images in a measurable and interpretable way, offering a principled approach to frequency-aware image synthesis.

## D  HYPERPARAMETERS DETAILS AND PERFORMANCES

**Training Configurations and Computational Cost**    All models were trained on NVIDIA A100 GPUs. VAE fine-tuning was performed for 60K steps with a batch size of 4 and a learning rate of $1 \times 10^{-5}3$. Table 9 details the specific training requirements for each backbone architecture.

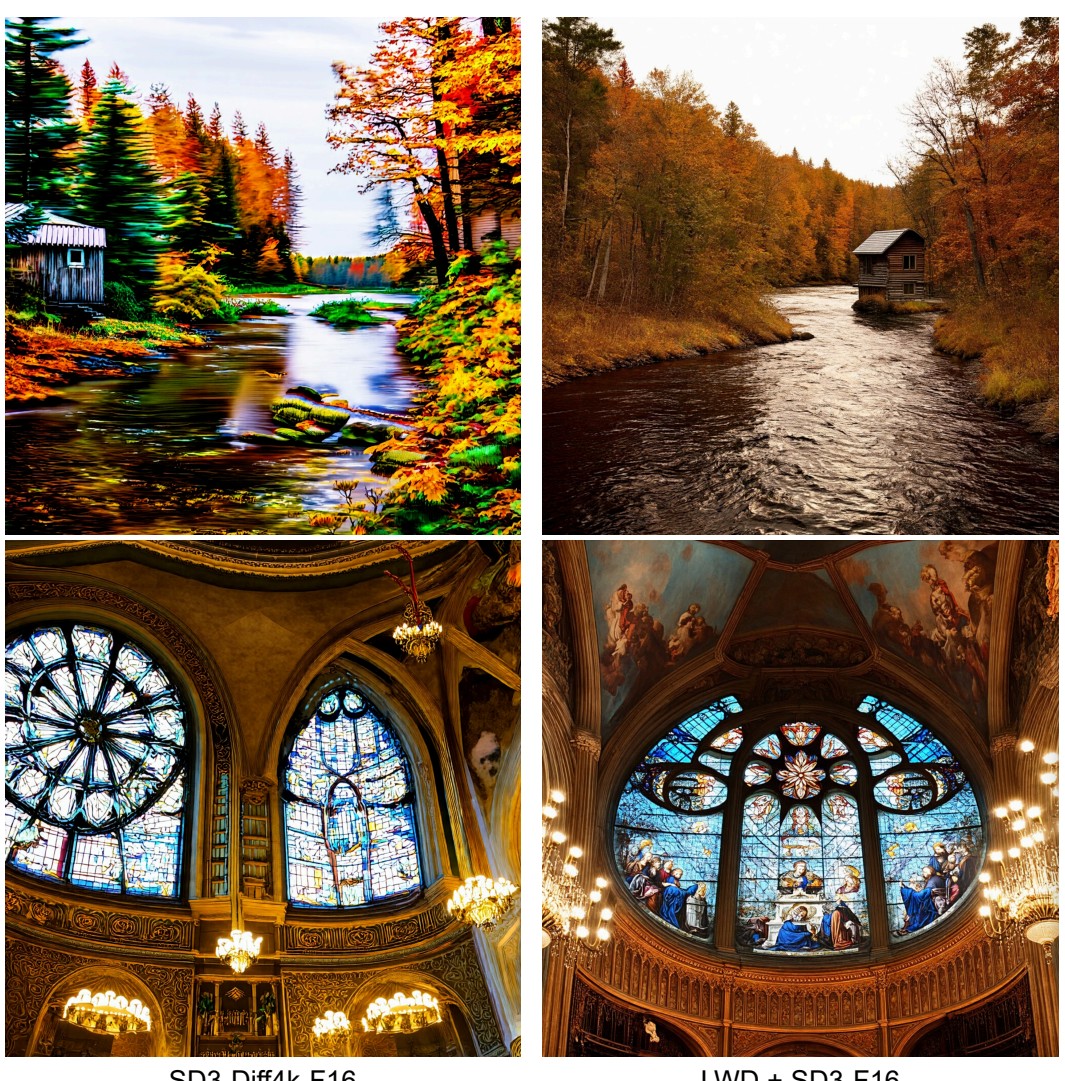

| SD3-Diff4k-F16 | LWD + SD3-F16 |

Figure 13: 2K generation of SD3-Diff4k-F16 vs LWD + SD3-F16.
Upper caption: *"A serene landscape features a winding river, flanked by trees with autumn foliage, leading to a rustic wooden cabin with a corrugated roof, set against a softly blurred background."*.
Lower caption: *"A grand interior featuring intricate stained glass windows, an elaborate rose window, ornate frescoes depicting biblical scenes, and elegant chandeliers illuminating the richly decorated walls and arches."*

Table 8: Comparison of frequency-sensitive metrics across different methods on Aesthetic-4K (Zhang et al., 2025) validation set.

| Metric | HLFR | RDFR ↓ | WQS ↑ | HFE | HFEI ↓ | FSIM ↑ | MS-SSIM ↑ |
|---|---|---|---|---|---|---|---|
| Real | 0.0560 | 0.0000 | 1.0000 | 0.0140 | 0.0000 | 1.0000 | 1.0000 |
| Sana-1.6B (Xie et al., 2025) | 0.0784 | 0.0558 | 0.4673 | 0.0196 | 0.6108 | 0.6128 | 0.1324 |
| LWD + Sana-1.6B | **0.0610** | **0.0537** | **0.4701** | **0.0144** | **0.5227** | **0.6217** | 0.1324 |
| SD3-Diff4k-F16 (Zhang et al., 2025) | 0.0691 | 0.0470 | 0.4624 | 0.0158 | 0.8064 | 0.6155 | 0.1296 |
| LWD + SD3-F16 | **0.0555** | **0.0437** | **0.4735** | **0.0144** | **0.4826** | **0.6245** | **0.1521** |
| PixArt-Sigma-XL | 0.0550 | **0.0409** | **0.4763** | 0.0119 | 0.6296 | 0.1354 | 0.4255 |
| LWD + PixArt-Sigma-XL | **0.0564** | 0.0500 | 0.4730 | **0.0150** | **0.6239** | **0.1478** | **0.5094** |

Table 9: Training configurations and efficiency gains for LWD across different backbones.

| Backbone | Res. | Batch Size | Iterations | Training Time |
|---|---|---|---|---|
| LWD + URAE (Flux) | 2048 | 1 | 2k | ∼4 hours |
| LWD + URAE (Flux) | 4096 | 1 | 2k | ∼24 hours |
| LWD + Diff4K (SD3) | 2048 | 8 | 10k | ∼48 hours |
| LWD + SANA | 2048/4096 | 2/1 | 33k | ∼24 hours |
| LWD + PixArt-$\Sigma$ | 2048 | 2 | 1.5k | ∼24 hours |

**Hyperparameters** All experiments were conducted on a system with 4 NVIDIA A100 GPUs. Our VAE fine-tuning objective (Equation 1) balances four terms. The weights were adopted from prior work (Wu et al., 2023), which provides extensive validation for these values. Following established practice, we set the weights to $\alpha = 0.25$, $\beta = 0.001$, and $\lambda = 0.05$.

**Training Overhead** LWD introduces a marginal overhead during training. The minor increase in peak GPU memory usage (Table 10) is due to the storage of intermediate tensors for the wavelet transform and energy masks. These tensors are small (the size of the latent map) and their memory footprint is insignificant compared to the large diffusion model backbone.

Table 10: Computational Overhead Analysis during training on a single A100 GPU (64GB).

| Method | Mem. Usage (%) | Mem. Usage (GB) | Time per 20 Steps (s) |
|---|---|---|---|
| Sana | 90.5 | 57.9 | ∼47 |
| Sana + LWD | 93.9 | 60.1 | ∼47 |

Another key advantage of LWD is its efficiency. It is a training-only strategy that requires zero architectural modifications. Consequently, an LWD-enhanced model has the exact same number of parameters and identical inference time as its baseline counterpart.

