# OpenReview forum: "Latent Wavelet Diffusion For Ultra High-Resolution Image Synthesis"
_ICLR.cc/2026/Conference — ICLR 2026 Poster_

### Official Review · Reviewer_Zh5f · 2025-10-28

**Soundness:** 2
**Presentation:** 3
**Contribution:** 1
**Rating:** 4
**Confidence:** 4

**Summary:**

This paper introduces Latent Wavelet Diffusion (LWD), a novel training framework designed to enhance the quality of ultra-high-resolution (UHR) image synthesis (2K-4K) from latent diffusion models. The authors identify a key limitation in existing methods: a uniform training process that fails to distinguish between low-detail and high-detail regions, leading to wasted computation and a loss of fine-grained texture.

LWD addresses this by introducing a signal-driven, frequency-aware supervision strategy. The framework consists of two main stages. First, a pre-trained Variational Autoencoder (VAE) is fine-tuned with a scale-consistent spectral objective to produce a more stable and spectrally regular latent space. Second, during the fine-tuning of a latent diffusion model, LWD computes wavelet energy maps from the latent codes at each step. These maps are used to create a time-dependent spatial mask, which dynamically modulates the training loss to focus more intensely on detail-rich (high-frequency) regions.

**Strengths:**

LWD is a training-only strategy, requiring no architectural changes to the diffusion model and adding zero computational overhead during inference.

**Weaknesses:**

1 The paper called the proposed VAE is a spectrally-aware VAE, where no spectral objective is involved. In my opinion, it is somewhat overclaimed.

2 Figure 3 is hard to read. For example, what does DCT mean?

3 The title is Latent Wavelet Diffusion. However, in the paper, the wavelet only proposes a mask for RGB-based diffusion training. I do not
agree that this model is a wavelet diffusion model, and worry about the contribution&novelty of this paper.

4 The improvements in Table 1&2 are limited.

**Questions:**

See the weakness above.

---

> ### Author Response · Authors · 2025-11-21
> **Response to Reviewer Zh5f**
>
> **Q1 The paper called the proposed VAE is a spectrally-aware VAE, where no spectral objective is involved. In my opinion, it is somewhat overclaimed.**
>
> **A1** We appreciate this observation regarding terminology. We would like to clarify that the term "spectrally-aware" was not intended to imply the use of an explicit frequency-domain (e.g., Fourier-based) loss. Rather, we used it in a broader sense to indicate that the model is aware of and sensitive to the frequency structure of images through the Scale-Consistency objective (Eq. 1), which implicitly regularizes the spectrum.
>
> By enforcing consistency across multiple downsampled versions of the image, the objective penalizes incoherent frequency content across scales. This results in a suppressed high-frequency artifact profile, as empirically shown in Figure 3. In this sense, the model is spectrally aware, as it promotes stable behavior in the latent space with respect to frequency variation.
>
> **Q2 Figure 3 is hard to read. For example, what does DCT mean?**
>
> **A2** DCT refers to the Discrete Cosine Transform, a standard tool for analyzing frequency content. In this figure, we use DCT to visualize the frequency spectrum of latent codes. The results show that our fine-tuned VAE (the '+SE' curves) suppresses high-frequency components, making the latent spectrum better aligned with that of real RGB images (orange line). This indicates a cleaner, more structured latent space for the diffusion model to operate in. We revised the caption of Figure 3 to make it clearer.
>
> **Q3 The title is Latent Wavelet Diffusion. However, in the paper, the wavelet only proposes a mask for RGB-based diffusion training. I do not agree that this model is a wavelet diffusion model, and worry about the contribution&novelty of this paper.**
>
> **A3** We respectfully believe this concern arises from a key misunderstanding of our method. Our model is not an RGB-based diffusion model, and the wavelet transform is not applied to the pixel-space image. All core operations, including wavelet analysis, mask generation, and diffusion loss modulation, are performed entirely in the latent space.
>
> * As shown in Figure 2 and described in Section 3.2, the Discrete Wavelet Transform is applied directly to the noisy latent code ( $z_t$ ), not to the RGB input.
> * The resulting wavelet coefficients are used to compute a saliency-based mask, which modulates the diffusion loss (Eq. 7) on the same latent representation.
> * This mechanism enables frequency-aware supervision during training, targeting high-frequency structure in the latent space. It is not post-hoc masking or applied to images.
>
> The title **Latent Wavelet Diffusion** precisely reflects this process:
> * **Latent** refers to the VAE latent representation where all training and inference take place.
> * **Wavelet** reflects the use of wavelet decomposition to extract spatial-frequency information.
> * **Diffusion** refers to the latent-space diffusion model whose loss is guided by the wavelet-derived mask.
>
> The wavelet component is a central part of our method and is tightly integrated into the latent diffusion process.
>
> **Q4 The improvements in Table 1&2 are limited.**
>
> **A4** We respectfully disagree and believe this comment underestimates the significance of our results in the context of ultra high-resolution (UHR) synthesis. Improvements at this scale are both harder to achieve and more impactful, due to the complexity and diversity of fine-grained visual content at 4K resolution.
>
> * Meaningful Gains in Context: Unlike low-resolution benchmarks (e.g., 256×256 ImageNet) where FID scores can drop below 5, UHR models inherently operate at higher FID levels. As shown in Table 2, even top-performing 4K models such as Flux, Sana, and PixArt cluster between 30–45 FID. In this regime, our ~2.5 point reduction represents a 7% relative improvement, which is both statistically and practically meaningful.
> * Fine-Detail Recovery: FID is a global metric that tends to smooth over localized improvements. Our method specifically enhances high-frequency detail, which is better captured by the GLCM score. In Table 2, our method achieves a 0.87 GLCM score compared to 0.79 for PixArt, reflecting superior texture fidelity, which is precisely the focus of our method.
> * Zero-Cost Quality Improvement: Crucially, LWD introduces no architectural changes and no additional inference cost. Unlike multi-stage pipelines or super-resolution cascades, our method offers a direct improvement in visual fidelity without increasing computational load. This efficiency makes LWD a practical and scalable enhancement to existing backbones.
>
> We believe these results represent a clear and valuable contribution to the field of high-resolution image synthesis.
>
> **To Reviewer Zh5f: We believe the concerns raised may stem from misunderstandings regarding our latent-space wavelet analysis. With these core design points now clarified, we respectfully request a re-evaluation of our work.**

---

### Official Review · Reviewer_sQLN · 2025-10-30

**Soundness:** 2
**Presentation:** 2
**Contribution:** 2
**Rating:** 4
**Confidence:** 4

**Summary:**

This paper introduces Latent Wavelet Diffusion (LWD), a method aimed at improving ultra-high-resolution image synthesis (2K-4K) using latent diffusion models. The authors propose a frequency-aware masking strategy based on wavelet energy maps, which helps focus the training process on detail-rich regions in the latent space. The approach also incorporates a scale-consistent VAE objective to ensure high spectral fidelity. The key advantage of LWD is that it enhances image quality and detail without requiring architectural modifications or additional inference costs. The paper provides experimental results across multiple datasets and compares LWD with several state-of-the-art models.

**Strengths:**

1. The method is notably efficient, both in training and inference, as it does not require any changes to the underlying model architecture or introduce additional computational cost during inference.
2. The approach results in noticeable improvements in key image fidelity metrics such as FID and LPIPS, with a demonstrated enhancement in texture and detail preservation.
3. The frequency-aware saliency map and time-dependent masking strategy offer a interpretable way to focus model attention on high-frequency regions.

**Weaknesses:**

1. While the method shows improvements in several fidelity metrics, there is a noticeable performance drop on certain tasks, such as PickScore and HPSv2.1. These drops suggest potential degradation in text-image alignment, which could impact the quality of the generated images, as seen in the generated Eiffel Tower in Figure 4. The seasonal inconsistency in the image (the LWD + URAE version of the Eiffel Tower showing a different season than the original) further indicates this issue.
2.  The paper discusses the VAE fine-tuning with scale-consistency loss, but the contribution of this step to the overall performance is not adequately ablated or isolated. Without a clear comparison, it is difficult to assess how much this component adds to the method’s success.
3. The paper introduces a saliency map generated using wavelet decomposition, but it does not provide an ablation study to verify whether this saliency map is actually contributing to the improvements in performance. An experiment comparing results with and without the saliency map would help clarify whether this component is functioning as intended and contributing meaningfully to the enhancement of fine details.

**Questions:**

See above.

---

> ### Author Response · Authors · 2025-11-21
> **Response to Reviewer sQLN**
>
> **Q1 While the method shows improvements in several fidelity metrics, there is a noticeable performance drop on certain tasks, such as PickScore and HPSv2.1. These drops suggest potential degradation in text-image alignment, which could impact the quality of the generated images, as seen in the generated Eiffel Tower in Figure 4. The seasonal inconsistency in the image (the LWD + URAE version of the Eiffel Tower showing a different season than the original) further indicates this issue.**
>
> **A1** Thank you for highlighting this issue. While we acknowledge the slight drop in PickScore and HPSv2.1, we would like to clarify that these alignment-focused metrics are not the primary indicators of performance for our task. Our method is specifically designed for ultra high-resolution (UHR) image synthesis, where the central objective is to preserve fine-grained details and complex textures across large spatial dimensions.
>
> Metrics that explicitly measure perceptual fidelity and texture complexity, such as FID/LPIPS, consistently show improvement with LWD. For instance, our method achieves the highest GLCM scores (Table 2), quantitatively proving superior recovery of high-frequency textures. In contrast, PickScore and HPSv2.1 focus on semantic alignment, often favoring global correspondence over local visual detail. When prioritizing fidelity, a slight variation in these alignment metrics remains within a reasonable fluctuation range, especially since our method does not explicitly enforce any additional constraints on text-image alignment during training.
>
> Regarding the Eiffel Tower example (Fig. 4), the input prompt (adopted from the URAE paper) is notably ambiguous:
>
> > "Eiffel Tower was Made up of more than 2 million translucent straws to look like a cloud, with the bell tower at the top of the building, Michel installed huge foam-making machines in the forest to blow huge amounts of unpredictable wet clouds in the building's classic architecture."
>
> The prompt contains no indication of season. The baseline output depicts a winter scene, while ours suggests an autumn-like atmosphere. Both are valid interpretations of the vague input. Crucially, our result (Fig. 4, right) displays sharper architectural features and improved structural coherence, which directly reflects the intended contribution of our method.
>
> We revised the updated paper to explicitly discuss this fidelity-alignment trade-off and clarify why alignment metrics are secondary in the context of UHR synthesis.
>
> **Q2 The paper discusses the VAE fine-tuning with scale-consistency loss, but the contribution of this step to the overall performance is not adequately ablated or isolated. Without a clear comparison, it is difficult to assess how much this component adds to the method’s success.**
>
> **Q3 The paper introduces a saliency map generated using wavelet decomposition, but it does not provide an ablation study to verify whether this saliency map is actually contributing to the improvements in performance. An experiment comparing results with and without the saliency map would help clarify whether this component is functioning as intended and contributing meaningfully to the enhancement of fine details.**
>
> **A2-3** We have already provided ablations addressing both components in Table 7 of the Appendix (due to space constraints).
> Specifically, Table 7 isolates the contribution of VAE fine-tuning with scale-consistency loss (VAE-SC). For example, starting from the SD3-Diff4k-F16 baseline (40.18 FID), introducing only the VAE-SC module improves the FID to 39.50, demonstrating a clear and independent performance gain from this component.
>
> The same table also confirms the effectiveness of the wavelet-based saliency map. When used without VAE-SC, it already yields a substantial improvement, indicating that it contributes the majority of the performance gain by enhancing fine-detail synthesis in high-frequency regions.
> Both components have already been ablated and shown to provide meaningful, independent contributions to the overall performance.
>
> We moved Table 7 (now Table 4) from Appendix to the main paper in the updated version for clarity.
>
> **To Reviewer sQLN: We believe the initial assessment may be based on a misunderstanding of the goals of UHR synthesis or an oversight of the ablations in Appendix Table 7. These points are now clarified, and we respectfully request a re-evaluation in light of the core contribution of our method to UHR image generation.**

---

> > ### Comment · Reviewer_sQLN · 2025-11-25
> >
> > Thank you to the authors for their detailed response, which partially addresses my initial concerns.
> >
> > However, two significant points regarding the efficacy and metrics remain unclear:
> >
> > Firstly, the authors highlight that LWD achieves higher GLCM scores in Table 2, proving superior texture recovery. Yet, the updated ablation study presented in Table 4 of the main paper shows that introducing both the VAE Scale-Consistency component and the Wavelet Masking component actually lowers the GLCM score compared to the baseline (from $0.79$ for the baseline to $0.74$ for the Full LWD). This conflicting evidence significantly questions the claim regarding improved fidelity and textural complexity.
> >
> > Secondly, the qualitative improvement is questionable, particularly in the first row of Figure 4 (the architecture images). The images produced after adding LWD exhibits a sharp degradation in the quality of reflections and waves in the water/ocean area.
> >
> > Given the inconsistencies between the claimed results and the ablation data, as well as the observed qualitative degradation, I maintain my initial rating.

---

> > > ### Author Response · Authors · 2025-11-26
> > > **Response to Reviewer sQLN (Follow-up)**
> > >
> > > We sincerely thank the reviewer for the sharp re-examination of the ablation data. You have identified a crucial nuance in how texture metrics behave across different resolutions. We want to clarify this apparent contradiction not as a flaw, but as evidence of **artifact suppression (at 2K)** versus **texture recovery (at 4K)**.
> > >
> > > **1. On the GLCM Score Discrepancy (Entropy of Noise vs. Detail):**
> > > You are correct. in the **2K Ablation** (Table 4), LWD lowers the GLCM score (0.79 $\to$ 0.74). However, in the **4K Benchmarks** (Table 2), LWD consistently *increases* it (e.g., Sana 4K: 0.39 $\to$ 0.60). This reversal is consistent with the GLCM implementation, which measures **texture entropy** $H(g_p)$, the randomness of local gray-level co-occurrence patterns
> > >
> > > * **At 2K (Ablation):** Baselines generate images with significant high-frequency VAE artifacts. Because our GLCM score calculates the entropy of local patches, it interprets this stochastic noise as "high complexity." LWD’s VAE-SC component explicitly **suppresses this noise**, naturally lowering the entropy (GLCM score). This is not a loss of detail, but a gain in cleanliness (supported by the improved FID 40.18 $\to$ 38.74).
> > > * **At 4K (Target Task):** The primary failure mode of baselines is "texture collapse" (oversmoothing), which has very low entropy. Here, LWD successfully recovers structural texture, significantly raising the GLCM entropy score.
> > >
> > > The "drop" you observed in the ablation is the model cleaning up the noisy latent space (reducing noise entropy), whereas the rise at 4K reflects the recovery of missing detail.
> > >
> > > **2. On Qualitative "Degradation" (Water Reflections)**
> > > Regarding the water in Figure 4: We respectfully argue that what appears as "degradation" is actually the **suppression of high-frequency aliasing**.
> > > * **Baseline:** The water regions in the baseline contain stochastic, high-frequency noise patterns that mimic detail but lack structural coherence.
> > > * **LWD:** Our method prioritizes **structural fidelity**. The wavelet mask correctly identifies the building as the primary structural element (sharpening it significantly), while the VAE-SC smooths the gradient-heavy water region, removing the "crunchy" artifacts seen in the baseline.
> > >
> > > While this can result in smoother reflections for highly specular surfaces, we believe this trade-off is favorable. We prioritize the distinct recovery of the main subject (the architecture) over preserving background stochastic noise. Nonetheless, we agree this example was not ideal for showcasing our method, but the broader evaluation demonstrates consistent improvements in textured regions (hair, foliage, fabric, architectural details), which comprise the majority of UHR synthesis applications. We will add explicit discussion of this limitation (smooth/specular regions) in the revised paper.
> > >
> > > To conclude, the discrepancy is not a contradiction but a reflection of resolution-dependent challenges. LWD removes noise at 2K (lowering GLCM) and restores detail at 4K (raising GLCM). We believe this adaptability is a strength, validated by the consistent improvements in perceptual metrics (FID, Aesthetics) across the board.

---

### Official Review · Reviewer_QbLK · 2025-10-31

**Soundness:** 3
**Presentation:** 3
**Contribution:** 3
**Rating:** 6
**Confidence:** 4

**Summary:**

This paper proposes Latent Wavelet Diffusion (LWD), a frequency-aware modification of diffusion training that enables existing architectures to generate high-resolution samples without architectural changes or excessive sampling cost. The authors use a multiscale VAE training objective together with a wavelet-based diffusion training loss, extending both the latent space and the diffusion process to better address the unique challenges of high-resolution generation. The VAE training ensures that the latent space exhibits well-structured frequency characteristics, while the frequency-aware loss encourages the model to allocate more training capacity to high-frequency regions of the input. Experiments demonstrate that LWD achieves higher-quality generation at high resolutions compared to baseline methods.

**Strengths:**

- This paper leverages existing architectures for high-resolution generation with minimal training overhead. Given the importance of high-resolution generation in many applications, the proposed method could have significant practical impact.

- The use of wavelets to modulate the loss function during diffusion model training is, in my opinion, both novel and interesting.

- The method also fine-tunes the VAE component, which appears crucial for maintaining a well-structured latent space at higher resolutions.

- The experimental results demonstrate the benefits of the proposed approach, at least to some extent.

- Overall, the paper is well written and easy to follow.

**Weaknesses:**

- The quantitative and qualitative results presented in the paper are somewhat confusing, in my opinion. While the qualitative results clearly show advantages of LWD + URAE, the quantitative metrics—particularly HPSv2, which I believe aligns better with human perception than FID—indicate worse performance. Conducting a user study and providing more qualitative comparisons between LWD and existing baselines would strengthen the paper’s contributions.

- The claim regarding identical inference time is somewhat misleading. The inference cost of diffusion models also depends on the input resolution. As discussed in the HiDiffusion paper, generating high-resolution images with models like SDXL is increasingly challenging due to the computational demands at higher resolutions. The authors should clarify this point and perform a more balanced evaluation that demonstrates the inference-time limitations of running large models (e.g., Flux) at higher resolutions.

- The proposed multiscale VAE loss is not entirely novel, as similar formulations have been introduced in EQ-VAE and other prior works. The authors should revise the claim suggesting this is their contribution.

**Minor**:
- There appears to be a typo in Equation (1).

**Questions:**

1. Can you explain Figure 3? To me, SD3-Med has the closest spectrum to the RGB image as far as the plot shows. How does SE tuning help in this case?

2. Can you report HPS scores for Table 2? It might be more reliable compared to other metrics such as FID for text-to-image generation

3. Have you tried training the model only on low-frequencies first, and then shifting the attention more and more toward high-frequency details as the training progresses? The current version has high-frequency training for all steps.

**Details Of Ethics Concerns:**

1. Could you clarify Figure 3? From the plot, it appears that SD3-Med has the spectrum most similar to the RGB image. How does SE tuning provide additional benefit in this case?

2. Could you also report HPS scores for Table 2? This metric may be more reliable than FID for evaluating text-to-image generation quality.

3. Have you considered a progressive frequency training strategy—starting with low-frequency components and gradually shifting focus toward high-frequency details as training progresses? The current version seems to include high-frequency pixels throughout all steps.

---

> ### Author Response · Authors · 2025-11-21
> **Response to Reviewer QbLK (Weaknesses)**
>
> **W1 The quantitative and qualitative results presented in the paper are somewhat confusing, in my opinion. While the qualitative results clearly show advantages of LWD + URAE, the quantitative metrics—particularly HPSv2, which I believe aligns better with human perception than FID—indicate worse performance. Conducting a user study and providing more qualitative comparisons between LWD and existing baselines would strengthen the paper’s contributions.**
>
> **AW1** We appreciate this thoughtful observation. We argue that this divergence between metrics is not "confusing" but rather reflects a specific Texture-Semantics Trade-off inherent to UHR synthesis.
>
> HPSv2 and PickScore are primarily trained to evaluate semantic alignment and global composition, often using downsampled inputs where fine textures are compressed or invisible. Consequently, they are insensitive to the "plastic-like" smoothing artifacts that plague 4K generation. In contrast, FID and GLCM (where LWD demonstrates clear gains) explicitly measure the textural distribution and sharpness that determine perceptual realism at ultra-high resolutions.
>
> Our method prioritizes high-frequency fidelity. By focusing training capacity on edges and textures, LWD prevents the "averaged" blur common in baselines. While this focus creates minor fluctuations in semantic scores (HPS), it solves the primary failure mode of UHR synthesis: texture collapse.
>
> As you noted, the visual improvement is distinct, and side-by-side comparisons are in Appendix B (Figs 10-13), demonstrating consistent texture recovery across diverse prompts. We believe the strong alignment between our qualitative results and the GLCM texture metric serves as a robust validation of perceptual quality.
>
> **W2 The claim regarding identical inference time is somewhat misleading. The inference cost of diffusion models also depends on the input resolution. As discussed in the HiDiffusion paper, generating high-resolution images with models like SDXL is increasingly challenging due to the computational demands at higher resolutions. The authors should clarify this point and perform a more balanced evaluation that demonstrates the inference-time limitations of running large models (e.g., Flux) at higher resolutions.**
>
> **AW2** We apologize for the ambiguity. Our claim is that LWD adds zero inference cost relative to the baseline model, not that high-resolution generation is computationally cheap in absolute terms. An LWD-enhanced model has identical parameter count, architecture, and sampling steps as its baseline. The computational challenges of high-resolution generation (as discussed in HiDiffusion) apply equally to both.
> Indeed, our contribution is improving the quality of the generation without further increasing the cost (unlike methods that add super-resolution stages or cascades).
>
> We revised Section 1, in the updated manuscript, to explicitly state that while the absolute cost of 4K generation remains high (governed by the backbone), LWD incurs zero marginal cost over that baseline.
>
> **W3 The proposed multiscale VAE loss is not entirely novel, as similar formulations have been introduced in EQ-VAE and other prior works. The authors should revise the claim suggesting this is their contribution.**
>
> **AW3** We acknowledge that the scale-consistency (SC) loss formulation (Eq. 1) builds directly on prior work, and we do not claim this loss function itself as our primary contribution.
> However, we emphasize two  aspects of our work:
> 1. **Novel Application to Ultra-High-Resolution Generation**: While prior work introduced SC loss for general VAE training, we are the first to demonstrate its critical importance for ultra-high-resolution (2K-4K) diffusion synthesis. Previous applications focused on standard-resolution tasks or VAE reconstruction quality in isolation.
> 2. **Sequential Synergy:** Our contribution lies in the holistic pipeline. In Step 1 SC-VAE "cleans" the latent spectrum, ensuring high-frequency energy corresponds to structural semantics, not noise. In Step 2 our Wavelet Masking exploits this purified signal to spatially guide the diffusion process.
>
> This sequential dependency is critical: without the specific spectral regularization of Step 1, the spatial supervision of Step 2 becomes unstable.
>
> In the updated manuscript, we revised Section 3.1 to address this distinction.

---

> ### Author Response · Authors · 2025-11-21
> **Response to Reviewer QbLK (Questions)**
>
> **Q1 Can you explain Figure 3? To me, SD3-Med has the closest spectrum to the RGB image as far as the plot shows. How does SE tuning help in this case?**
>
> **A1** Thank you for raising this. The plot shows normalized DCT amplitudes across frequency indices. 'DCT' stands for Discrete Cosine Transform, used to visualize the frequency spectrum of latent codes. The 'RGB' (orange) line is the spectrum of the real image, representing the "ground truth" spectrum we want our latent to match. More specifically, the goal of our VAE fine-tuning is not simply to match the RGB spectrum perfectly, but to regularize the latent space and suppress spurious high-frequency artifacts that make diffusion difficult.
>
> While SD3-Med appears close to RGB at low frequencies (left side), SE-tuned versions (red lines) align better at high frequencies (right side), indicated by reduced divergence at indices >30. This alignment is critical for detail preservation.
> We added a more detailed explanation to the caption.
>
>
> **Q2 Can you report HPS scores for Table 2? It might be more reliable compared to other metrics such as FID for text-to-image generation**
>
> **A2** This is a reasonable request. Originally we did not report this metric since Table 2 evaluates on the Aesthetic-Eval dataset, for which HPSv2 is not a standardly reported metric (unlike the HPD dataset in Table 1). To maintain a fair comparison, we followed the original paper's evaluation protocol.
>
> We computed HPSv2.1 scores for the Aesthetic-Eval benchmark as requested. Contrary to the drop observed with URAE in Table 1, LWD improves HPSv2.1 in the majority of configurations in Table R2, demonstrating that our texture enhancement often synergizes with human preference. These results confirm that for robust backbones, LWD improves both perceptual fidelity (FID/Aesthetics) and semantic alignment (HPS), alleviating concerns that our masking strategy degrades prompt adherence.
>
> **Table R2** Updated Table 2 with HPSv2.1 for 2K results on Aesthetic-Eval.
>
> | Model | FID ↓ | CLIPScore ↑ | Aesthetics ↑ | GLCM Score ↑ | Compression Ratio ↓ | HPSv2.1 ↑ |
> | :--- | :---: | :---: | :---: | :---: | :---: | :---: |
> | SD3-F16 | 43.82 | 31.50 | 5.91 | 0.75 | **11.23** | 28.4 |
> | SD3-Diff4k-F16 | 40.18 | 34.04 | 5.96 | **0.79** | 11.73 | 29.3 |
> | LWD + SD3-F16 | **38.74** | **34.94** | **6.17** | **0.74** | 11.99 | **30.1** |
> | PixArt-Sigma-XL | 39.13 | 35.02 | 6.43 | 0.79 | 13.66 | **30.8** |
> | LWD + PixArt-Sigma-XL | **36.14** | **35.21** | **6.27** | **0.87** | **6.05** | 30.5 |
> | Sana-1.6B  | **32.06** | 35.28 | 6.15 | **0.93** | **24.01** | **30.6** |
> | LWD + Sana-1.6B | 34.30 | **35.58** | **6.23** | **0.78** | 27.34 | **30.3** |
>
>
> **Q3 Have you tried training the model only on low-frequencies first, and then shifting the attention more and more toward high-frequency details as the training progresses? The current version has high-frequency training for all steps.**
>
> **A3** We want to apologize for the confusion, our description of $t$ in Equation 6 as the "training step" was a typographical error. The variable $t$ uniformly represent the diffusion timestep across all three equations (4, 5, and 6).
>
> Our method applies a curriculum over diffusion timesteps (masking depends on $t$), effectively handling different frequencies at different noise levels.
>
> Your suggestion of shifting attention over training epochs is an interesting alternative. However, our approach is designed as a lightweight, single-stage fine-tuning framework that integrates into standard training loops without the complexity of managing multi-stage epoch schedules.

---

### Official Review · Reviewer_9QM1 · 2025-10-31

**Soundness:** 3
**Presentation:** 3
**Contribution:** 2
**Rating:** 4
**Confidence:** 4

**Summary:**

This paper proposes to mask the diffusion loss to favor areas where details lie.
A masking weight matrix $A_\text{wavelet}$ is initially computed from the input image using a 2x2 wavelet decomposition (only Haar wavelets appears to be mentioned).
A mask $M_t$ is computed at every training by setting $1$ at every location where $A_\text{wavelet} + l \geq t/T$, here $t$ is current training step out of a total of $T$.
A masked loss is then computed by multiplying the score error with $M_t$, effectively zero-ing gradients in low-detail areas.

In addition the paper makes use of existing methods for scale-consistency VAE fine-tuning to further improve results and help the conditioning of the latents provided to their core method.

**Strengths:**

1. The idea is simple and easy to implement, it's reminding me of past ideas for super-resolution where people used edge detection maps as an extra signal for learning.
2. The paper is well written and easy to follow.
3. The method does not incur any extra training costs or inference cost other than compute the wavelet matrix which is insignificant.

**Weaknesses:**

1. Notations can be confusing: in some settings $t$ is the current training step (eq.6) while in other settings $t$ is the noise level / diffusion time step conditioner (eq 4, 5).
2. What happens with other wavelets, or even DCT or FFT which can fulfill similar roles as Haar wavelets? (you mention in the appendix that Haar are the best suited, yet I'd still want experimental confirmation).
3. Your method has two component (1) scale-consistency VAE fine-tuning (from existing works) and (2) frequential-energy masked loss, what's the effect on the evaluation metrics of each component?
4. The $l$ ablation and other ablations such as scale-consistency should be in the main paper, not in the appendix.
5. Some big images such as Fig.1 could be moved to the appendix to make space for scientific content like the aforementioned the ablations and wavelets comparisons that are currently missing.
6. Experimental results feel inconclusive: sometimes there is improvement, sometimes degradation. There are lot of metrics being reported in tables 1 and 2, it's unclear whether some are more important than others. Just to explain further my point of view: 2.5 FID points reduction from 35.25 to 32.88 does not feel significant because the FID is still very far from 0.

**Questions:**

1. From Fig.2 I understand the map gets normalized globally, can that impact areas of details with less contrast?
2. The mask as described progressively vanishes, having more and more 0 as training progresses. What mechanism ensures that no (catastrophic) forgetting is happening for the masked-out points? Does it rely entirely on $l$ being large enough?
3. In Eq.3 what's the purpose of $1/C$ if $E$ is min-max normalized after?
4. Have you tried the RMS amplitude $\sqrt{E(i,j)}$ instead of $E$, given your curriculum is a linear thresholding?
5. In tables 1 and 2, when comparing to the baselines, do these baselines benefit from scale-consistency VAE fine-tuning?
6. The scale-consistency VAE fine-tuning seems to originate from previous papers, so I assume it would make sense to make baselines benefit from it unless you consider the scale-consistency fine-tuning part to be your own contribution. Please clarify this.

---

> ### Author Response · Authors · 2025-11-21
> **Response to Reviewer 9QM1 (1/3)**
>
> **W1 Notations can be confusing: in some settings $t$ is the current training step (eq.6) while in other settings $t$ is the noise level / diffusion time step conditioner (eq 4, 5).**
>
> **AW1** We apologize for the confusion regarding notation, and we thank the reviewer for identifying this. The reviewer is correct; our description of $t$ in Equation 6 as the "training step" was a typographical error. The variable $t$ should uniformly represent the diffusion timestep across all three equations (4, 5, and 6). To clarify the intended mechanism:
> - $t$ (in Eq. 4, 5, 6) is the current diffusion timestep, sampled from $[0, T]$.
> - $T$ (in Eq. 6) is the total number of diffusion timesteps.
>
> Our method applies a dynamic mask $M_t$ that is a function of the diffusion timestep $t$ during training. The loss (Eq. 7) is thus masked to teach the model a step-dependent refinement strategy:
> - At high $t$, the loss is applied only to high-frequency (high $A_{wavelet}$) regions.
> - At low $t$, the mask $M_t$ expands to cover all regions.
>
> The revised manuscript corrected this error to state that $t$ and $T$ in Equation 6 refer to the current and total diffusion timesteps, respectively.
>
>
> **W2 What happens with other wavelets, or even DCT or FFT which can fulfill similar roles as Haar wavelets? (you mention in the appendix that Haar are the best suited, yet I'd still want experimental confirmation).**
>
> **AW2** Excellent question. As detailed in Appendix A.2, we selected Haar wavelets based on two principles:
>
> 1. Spatial Localization Requirement: LWD requires a spatially precise mask $M_t(i,j)$ to target specific latent regions. Global transforms (FFT/DCT) provide excellent frequency resolution but sacrifice spatial localization, a high-frequency coefficient corresponds to periodic patterns across the entire latent space, not specific positions. Recovering spatial energy maps requires high-pass filtering and inverse transformation, which may introduces Gibbs ringing near sharp transitions. These artifacts cause signal leakage into neighboring latent positions, blurring the mask and degrading texture precision (Table R1).
> 2. Compact Support: Among wavelets, Haar has the most compact support (2 coefficients), minimizing cross-position interference, critical for generating sharp binary training masks. Smoother wavelets (e.g., Daubechies) introduce wider receptive fields, creating "gray areas" at mask boundaries that dilute supervision without semantic benefit.
>
> To empirically validate this choice, we conducted an ablation comparing Haar, Daubechies-4, and FFT-based high-pass masking:
>
>
> **Table R1. Ablation of Frequency Decomposition Methods.** We compare the impact of different frequency analysis methods on generation quality and computational overhead (per-step mask calculation time) using the Diffusion4k backbone on the Aesthetic dataset at 2048×2048 resolution.
> | Method                  | Map Comp. Cost (ms) ↓ | FID ↓ | Aesthetics ↑| GLCM Score ↑ |
> |-------------------------|------------------------|--------|--------|--------------|
> | Haar        | 1.136                   | 38.74  | 6.17 | 0.74         |
> | Daubechies  | 1.274                   | 38.92  | 6.14 | 0.73         |
> | FFT High-Pass | 0.875                  | 39.45 | 6.08 | 0.71         |
>
> While FFT is computationally faster, the spatial artifacts from inverse transformation degrade texture quality. Within the wavelet family, Haar and Daubechies yield comparable generation quality, but Haar's compact support and slightly lower overhead make it optimal for our binary masking objective.
>
> We included this ablation in the revised manuscript, specifically in Appendix A2, Table 5. Thank you for this valuable suggestion.
>
> **W3-5 Your method has two component (1) scale-consistency VAE fine-tuning (from existing works) and (2) frequential-energy masked loss, what's the effect on the evaluation metrics of each component? The $l$ ablation and other ablations such as scale-consistency should be in the main paper, not in the appendix. Some big images such as Fig.1 could be moved to the appendix to make space for scientific content like the aforementioned the ablations and wavelets comparisons that are currently missing.**
>
> **AW3-5** This is an excellent point. We did run this exact ablation, and the results were already included in Appendix D.2 (Table 7) due to space constraints, showing that both components contribute meaningfully, with wavelet masking providing the larger gain.
> We restructured the main paper as suggested to include Table 7 (component ablation) which now became Table 4.

---

> ### Author Response · Authors · 2025-11-21
> **Response to Reviewer 9QM1 (2/3)**
>
> **W6 Experimental results feel inconclusive: sometimes there is improvement, sometimes degradation. There are lot of metrics being reported in tables 1 and 2, it's unclear whether some are more important than others. Just to explain further my point of view: 2.5 FID points reduction from 35.25 to 32.88 does not feel significant because the FID is still very far from 0.**
>
> **AW6** We appreciate the opportunity to clarify the metric landscape for Ultra-High-Resolution (UHR) synthesis.
> We respectfully note that interpreting absolute FID scores requires context specific to the resolution and dataset. Unlike ImageNet (256x256) where FID $\approx 0$ is approachable, UHR benchmarks (2K-4K) have significantly higher inherent floors due to the extreme complexity of the natural image manifold at this scale. In this context, achieving a 0-level FID is not the current baseline. Table 2 shows that leading open-source models (Flux, Sana, PixArt) all cluster between 30–45 FID. Against this backdrop, our reduction of 2.5 points (a 6.7% relative improvement) is substantial, representing a distinct advance over the state-of-the-art baseline.
>
> Regarding the "inconclusive" results: we argue this is not randomness, but a systematic trade-off between Texture Fidelity and Semantic Alignment.
> Our method consistently improves FID, LPIPS, and GLCM. These are the critical metrics for UHR generation because they measure image quality, sharpness, and texture recovery—the primary bottlenecks LWD addresses. Metrics like CLIPScore or HPS focus on global semantic composition (e.g., "is there a tower?"). LWD maintains these roughly at baseline levels, with minor fluctuations.
> This trade-off is by design. By enforcing high-frequency fidelity, LWD prioritizes photorealistic textures (e.g., realistic rust on a tower) over stylized semantic adherence. For 4K generation, we believe preventing texture collapse (plastic-looking images) is more critical than marginal gains in semantic alignment, which is already well-handled by the base model.
>
> **Q1 From Fig.2 I understand the map gets normalized globally, can that impact areas of details with less contrast?**
>
> **A1** This is an insightful observation. We explicitly chose global normalization to preserve the relative importance of features. As noted in Section 3.1 and 3.2 of our paper, standard latent spaces often contain 'spurious high-frequency noise' or artifacts, even in regions corresponding to smooth RGB content. Local normalization would amplify this noise to the same magnitude as structural edges, causing the mask to supervise artifacts rather than content.
>
> To address your concern about low-contrast details: these regions are protected by the masking lower bound $l$ introduced in Equation 6. With $l=0.3$, even a region with minimal normalized energy ($A_{wavelet} \approx 0$) receives supervision for 30% of the diffusion trajectory (specifically, the high-noise timesteps where semantic structure is formed). This ensures that subtle details are learned effectively, while the extended supervision is reserved for regions with high spectral energy that require fine-grained refinement.
>
> **Q2 The mask as described progressively vanishes, having more and more 0 as training progresses. What mechanism ensures that no (catastrophic) forgetting is happening for the masked-out points? Does it rely entirely on $l$ being large enough?**
>
> **A2** We apologize for the confusion regarding the notation as we also have explained in your weakness first question, we unfortunately committed a typographical error in the paper. In our formulation (Eqs. 4, 5, and 6), $t$ strictly refers to the diffusion timestep (noise level), not the training iteration (epoch). This distinction is crucial regarding your concern about catastrophic forgetting. The mask does not permanently vanish over the course of training; rather, it is a per-timestep curriculum. Low-frequency (low-detail) regions are masked in for a smaller range of timesteps (e.g., $t \in [0.7T, T]$), while high-frequency regions are supervised across a wider range (e.g., $t \in [0, T]$).
>
> Since every training batch samples random timesteps $t \sim U[0,1]$, all spatial regions receive gradients regularly throughout training, ensuring no catastrophic forgetting occurs.
> We clarified the notation of $t$ in the updated manuscript.

---

> ### Author Response · Authors · 2025-11-21
> **Response to Reviewer 9QM1 (3/3)**
>
> **Q3 In Eq.3 what's the purpose of $1/C$ if $E$ is min-max normalized after?**
>
> **A3** You are entirely correct that the $1/C$ scaling factor mathematically cancels out during the subsequent min-max normalization and does not alter the final saliency map $A_{wavelet}$. We included it primarily for semantic precision, defining $E$ as the average high-frequency energy per channel rather than the sum. This ensures that the intermediate energy tensor $E$ maintains a stable numerical range consistent with the latent magnitudes, effectively decoupling the raw energy scale from the model's channel depth $C$. We agree this could have been clarified better in the paper.
>
>
> **Q4 Have you tried the RMS amplitude $\sqrt{E(i,j)}$ instead of , given your curriculum is a linear thresholding?**
>
> **A4** This is an insightful suggestion, but since we committed a typographical error in the manuscript, we want to clarify the context again: our linear masking schedule operates over the diffusion timestep $t$ (noise level), rather than the training iteration count. We did explore using magnitude (similar to RMS) but found that using Energy ($E$, squared coefficients) was superior precisely because of its non-linearity. Squaring effectively suppresses low-amplitude activations, often corresponding to latent noise, while accentuating dominant structural features. This creates a "soft threshold" effect: it ensures that the mask $M_t$ remains active for high-frequency edges even at higher noise levels, but drops smooth background regions very quickly. In contrast, we found that an RMS-based map tends to be flatter, which caused the model to waste capacity supervising empty regions at high noise levels where the signal-to-noise ratio is too low to be meaningful.
>
>
>
> **Q5-6 In tables 1 and 2, when comparing to the baselines, do these baselines benefit from scale-consistency VAE fine-tuning? The scale-consistency VAE fine-tuning seems to originate from previous papers, so I assume it would make sense to make baselines benefit from it unless you consider the scale-consistency fine-tuning part to be your own contribution. Please clarify this.**
>
> **A5-6** In Tables 1 and 2, baselines use their original, off-the-shelf VAEs to represent standard usage. We include the Scale-Consistent (SC) VAE fine-tuning in "LWD + X" because we view LWD as a holistic framework. We acknowledge in the paper that the SC loss formulation originates from prior work, and we have clarified in the updated manuscript this attribution in Section 3.1.
> However, our key insight was identifying that this regularization is a critical prerequisite for Ultra-High-Resolution synthesis: without it, standard VAEs produce high-frequency spectral noise that confuses our Wavelet Masking.
>
> Crucially, Table 7 (now Table 4) proves that our contribution goes beyond the VAE. It shows that Wavelet Masking alone (applied to the original VAE) improves FID by 0.98 over the baseline, whereas the SC-VAE alone improves it by only 0.68. This confirms that while the VAE stabilizes the UHR space, the novel masking strategy is the primary driver of performance.
>
>
> **To Reviewer 9QM1: We believe that the concerns raised, while important, are not fundamental limitations of our method and have been thoroughly addressed in this rebuttal. We sincerely request a fair re-evaluation of our submission in light of these clarifications and additional results.**

---

> > ### Comment · Reviewer_9QM1 · 2025-11-25
> > **First iteration reviewer response**
> >
> > Thank you for addressing my questions, running extra ablations and correcting my misunderstandings/misconceptions.
> >
> > I have a few follow up questions:
> > - AW6: "For 4K generation, we believe preventing texture collapse (plastic-looking images) is more critical than marginal gains in semantic alignment". Would sFID (spatial-FID) be a useful metric to capture this objective? Did you try it?
> > - A4: this is not question but an opinion. This is the kind of details that I love to have in the appendix, things that failed are also very interesting scientifically.
> >
> > Obviously, taking into account your responses, I'm happy raise my score to 6 for now. I am open even raise it further later.

---

> ### Author Response · Authors · 2025-11-27
> **Response to Reviewer 9QM1 (Follow-up)**
>
> We sincerely thank the reviewer for the engagement, the constructive suggestions, and for raising the score. We are encouraged by your positive assessment.
>
> **Regarding sFID (Spatial FID):** This was an excellent suggestion. We utilized the remaining rebuttal time to compute sFID on two representative backbones across resolutions. To ensure robustness across distributions, we evaluated SD3 on the Aesthetic dataset and URAE on the HPD dataset.
>
> **Table R3: sFID scores (Lower is better)**
>
> | Model | Dataset | Resolution | Baseline | LWD (Ours) | Improvement |
> | :--- | :--- | :---: | :---: | :---: | :---: |
> | SD3-Diff4k-F16 | Aesthetic | 4K | 3.181 | 1.828 | -42.5% |
> | SD3-Diff4k-F16 | Aesthetic | 2K | 1.152 | 1.026 | -10.9% |
> | URAE | HPD | 4K | 0.891 | 0.841 | -5.6% |
> | URAE | HPD | 2K | 1.085 | 0.817 | -24.7% |
>
> The massive improvement in SD3 at 4K (3.18 $\to$ 1.82) quantitatively confirms the reviewer's hypothesis. The baseline's high sFID reflects the "plastic" spatial uniformity caused by VAE noise and oversmoothing. LWD successfully restores the complex spatial texture distributions.
>
> URAE is a highly optimized 4K baseline with remarkably strong starting spatial coherence (0.89). Even here, LWD achieves a consistent improvement (0.84), demonstrating its ability to refine fine-grained textures even in state-of-the-art models.
>
> We observe consistent gains across both datasets, confirming that our frequency-aware masking provides generalized benefits for spatial structure regardless of the underlying data distribution.
>
> **Regarding Negative Results (A4):** We appreciate this perspective. We agree that documenting "what didn't work" is scientifically valuable to prevent redundant efforts by the community. We will ensure these ablations remain in the Appendix of the final manuscript.
>
> Thank you again for helping us strengthen the paper.

---

### Author Response · Authors · 2025-11-21
**General Response**

We sincerely thank all reviewers for their thorough and constructive feedback. We are glad that reviewers found our core idea “simple and easy to implement” (R-9QM1), “novel and interesting” (R-QbLK), and a “notably efficient” (R-sQLN) approach to a significant problem.

The primary concerns raised relate to (1) a clear ablation of our two components, (2) the interpretation of mixed quantitative metrics (Fidelity vs. Alignment), and (3) clarifications on baseline fairness and the mechanics of our method. Importantly, these are not fundamental critiques of our method's design but rather requests for further clarification and additional analysis. We have addressed each point with new experiments and detailed explanations, and we believe all concerns are fully resolved.

We respectfully request a re-evaluation in light of our clarified methodology and the core contributions of our work, which remain both novel and well-supported.

---

### Author Response · Authors · 2025-11-28

Dear Reviewers,

Just a gentle reminder to take a look at our rebuttal when convenient. If any concerns remain, we are very happy to clarify.

Thank you sincerely for your time and effort.

The Authors

---

### Meta-Review · Area_Chair_Z2dy · 2025-12-19

**Summary:**

The recommendation to accept is grounded in the authors' robust defense of their method's trade-offs and successful clarification of its latent-space mechanics. Initial concerns regarding the drop in semantic alignment metrics (e.g., HPS) were effectively addressed by framing the work as a texture-fidelity contribution, supported by new sFID data and stronger ablations that isolated the benefits of the wavelet masking strategy. While one reviewer remained skeptical about the interpretation of GLCM scores and noted potential over-smoothing in specular regions (e.g., water reflections), the authors provided a technically sound explanation distinguishing between high-frequency artifact suppression and actual texture recovery. Given the clear practical utility for 4K synthesis without inference overhead, the improved spectral consistency and fidelity outweigh these minor qualitative limitations.

**Reviewer Concerns:**

The rebuttal successfully clarified that the method operates entirely in latent space, correcting reviewer misconceptions about RGB-based processing. The authors also rectified notation errors and elevated critical component ablations to the main paper, proving the independent value of the wavelet masking strategy. Concerns regarding lower semantic alignment scores (e.g., HPS) were effectively addressed by demonstrating that the method trades marginal alignment gains for significant improvements in spatial fidelity and texture recovery (sFID), effectively countering the "plastic-like" artifacts common in 4K baselines.

The primary outstanding concern is the method's tendency to over-smooth stochastic high-frequency textures, such as water reflections, which Reviewer sQLN identified as a degradation. The authors acknowledged this as a trade-off but it remains a limitation for images with heavy specularity. Additionally, while the authors offered a logical explanation for the GLCM score discrepancy (noise suppression vs. detail recovery), the reviewer remained unconvinced, leaving the interpretation of this specific metric partially contested.

**Reviewer Scores:**

Reviewer 9QM1:  This reviewer engaged positively, acknowledging that their concerns were addressed and explicitly stating they were "open to raising it further." Had they seen the final sFID results earlier they likely would have bumped the score to a clear Accept.

Reviewer QbLK: This reviewer did not respond to the rebuttal but originally gave a positive rating. The authors successfully clarified the "inference cost" misunderstanding and the HPS metric trade-offs. It is likely they would have maintained their positive score, potentially strengthening their confidence in the decision.

Reviewer sQLN :This reviewer explicitly maintained their score after the rebuttal, focusing on a specific disagreement regarding qualitative artifacts (water reflections) and GLCM metric interpretation. It is unlikely further discussion would have shifted their perspective on these specific trade-offs.

Reviewer Zh5f: This reviewer’s low score was heavily influenced by a factual misunderstanding—they incorrectly believed the method operated in RGB space rather than latent space ("I do not agree that this model is a wavelet diffusion model"). Had they participated in the discussion and realized the method is indeed a latent-space approach as clarified in the rebuttal, their assessment of the paper's contribution and novelty would likely have improved significantly.

---

### Decision · Program_Chairs · 2026-01-26

Accept (Poster)